# Adaptive Sampling for Efficient Softmax Approximation

**Tavor Z. Baharav†**
Eric and Wendy Schmidt Center
Broad Institute
Cambridge, MA, 02142
`baharav@broadinstitute.org`

**Ryan Kang†**
Department of Computer Science
Stanford University
Stanford, CA 94305
`txryank@stanford.edu`

**Colin Sullivan†**
AI Division
Software Engineering Institute
Pittsburgh, PA 15213
`csullivan@sei.cmu.edu`

**Mo Tiwari**
Department of Computer Science
Stanford University
Stanford, CA, 94305
`motiwari@stanford.edu`

**Eric Luxenberg**
Gridmatic
Cupertino, CA 95014
`eric@gridmatic.com`

**David Tse**
Department of Electrical Engineering
Stanford University
Stanford, CA 94305
`dntse@stanford.edu`

**Mert Pilanci**
Department of Electrical Engineering
Stanford University
Stanford, CA 94305
`pilanci@stanford.edu`

## Abstract

The softmax function is ubiquitous in machine learning and optimization applications. Computing the full softmax evaluation of a matrix-vector product can be computationally expensive in high-dimensional settings. In many applications, however, it is sufficient to calculate only the top few outputs of the softmax function. In this work, we present an algorithm, dubbed `AdaptiveSoftmax`, that adaptively computes the top $k$ softmax values more efficiently than the full softmax computation, with probabilistic guarantees. We demonstrate the sample efficiency improvements afforded by `AdaptiveSoftmax` on real and synthetic data to corroborate our theoretical results. `AdaptiveSoftmax` yields $> 10x$ gain over full softmax computation on most datasets, yielding up to 30x improvement for Mistral7B evaluated on the Wikitext dataset. The adaptive method we propose for estimating the partition function (the softmax denominator) is of independent interest and can be used in other applications such as kernel density estimation.

## 1 Introduction

The softmax function appears in many different fields and applications. It is often used in multiclass classification problems, as the final operation in a neural network to obtain a probability distribution over classes, in reinforcement learning to obtain a probability distribution over possible actions, and in statistical mechanics to derive various thermodynamic quantities.

In machine learning applications, the softmax function often appears as the final operation in classification models and in attention layers. Crucially, the softmax function takes a vector of weights

---

†denotes equal contribution

38th Conference on Neural Information Processing Systems (NeurIPS 2024).

as input and returns a probability distribution defined by those weights. Formally, the softmax function for a given temperature parameter $\beta \in \mathbb{R}$ is defined as:

$$\sigma_\beta(\mu)_i = \frac{e^{\beta\mu_i}}{\sum_j e^{\beta\mu_j}}. \tag{1}$$

where $\mu \in \mathbb{R}^n$ is the input vector of weights, also referred to as the *logits*. Usually, the logits are the result of a matrix-vector product (e.g., in a fully connected layer where the softmax is used as the nonlinear activation function). The output of the softmax function (Equation 1) is a probability distribution that is a "soft" version of the max operator that is differentiable with respect to the logits. The softmax function can thus be used in gradient-based algorithms as a proxy for the non-differentiable max function.

Intuitively, the temperature parameter $\beta$ controls the peakiness of the softmax output. A larger $\beta$ corresponds to a peakier distribution and a "harder" max. The choice of $\beta = 1$ corresponds to the canonical softmax function $\sigma$, and the choice of $\beta = \infty$ corresponds to the "hard" argmax. The denominator of Equation (1), $\sum_j e^{\beta\mu_j}$, is called the partition function and is denoted by $Z_\beta$.

The softmax function is critical in many popular, recent machine learning applications like large language models (LLMs). However, it can present a computational bottleneck in high-dimensional applications. During the training of neural networks, for example, each training example $x$ requires the computation of the softmax function $\sigma_\beta(x)$, the partition function $Z_\beta(x)$, and their gradients. During inference of these models, the number of possible labels for next-token prediction corresponds to the vocabulary size, which can be in the hundreds of thousands for common languages such as English. As such, there has been significant recent interest in accelerating the computation of the softmax function and its derivatives [37, 14, 15].

**Key Observations:** In many applications, we are only interested in identifying the top few outputs of the softmax function; in these settings, it is unnecessary to compute the smaller entries. This suggests that some of the computation of the full softmax function may be unnecessary and motivates our study. First, we observe that when the input vector to the softmax function is the result of a matrix-vector product, we can approximate the intermediary computation instead of exactly computing it. This, in turn, allows us to approximate the output of the softmax function and converts the problem of computing the softmax function from a computational one to a statistical one. We also note that the softmax output is heavily influenced by the largest input elements which suggests that we can allocate computation adaptively to larger input elements to estimate them with greater certainty. This procedure is inspired by recent work in multi-armed bandits that converts computational problems to statistical ones [4].

**Outline:** We begin this study with a summary of related work in Section 2. In Section 3, we formalize the reduction of computing the softmax function to a statistical estimation problem. In Section 4, we propose the `AdaptiveSoftmax` algorithm based on this reduction. In the same section, we provide probably approximately correct (PAC) guarantees for `AdaptiveSoftmax` and prove that it is more efficient than brute force computation. Crucially, `AdaptiveSoftmax` allocates greater computational resources towards important output values. In Section 5, we demonstrate the empirical advantages of our algorithm in several real-world applications, including in a multiclass classification setting and in large language models. In Section 6, we conclude with a discussion of further applications, potential limitations, and directions for future work.

## 2   Related Work

Recent work has identified the computational complexity of the softmax function as a significant bottleneck in recent machine learning applications [37]. Well before the attention revolution, [36] proposed methods to accelerate softmax computation via a hierarchical model. In their work, a binary tree representing possible outputs as leaves is used and, at each step, the model must predict which path to traverse to a leaf. For a balanced tree with $n$ leaves, the computational complexity of the softmax function is reduced to $O(\log n)$ from $O(n)$, at the expense of $O(n)$ internal classifiers and providing only an approximate output dictated by the quality of the clustering. Google's word2vec models used Huffman trees instead of vanilla binary trees in a similar tree-based approach [37]. Other approaches include target sampling, noise contrastive estimation and self normalization (summarized [14]), but all of these methods reduce the complexity in terms of the vocabulary size $n$, rather than

by the dimension $d$. Additionally, our proposed algorithm provides direct PAC guarantees on the original softmax output, instead of approximating the softmax in a sequence of steps without provable accuracy guarantees. Independently, some works have developed fast methods to approximate the softmax gradients during training by using importance sampling over the classes, improving scaling with respect to $n$ [11, 10] leading to a sampled softmax. This is in contrast with `AdaptiveSoftmax`, which utilizes importance sampling in an orthogonal direction to subsample the features efficiently, enabling gains in $d$ at both train and test time. These sampled softmax methods were later specialized to kernel-based sampling, resulting in provably bounded bias [12, 38]. However, these and other optimized methods [23] typically require prior knowledge about the desired output label frequencies, leaving them susceptible to phenomena like distribution shift between training and inference data, where the frequency distribution changes at the time the model is evaluated. Unlike these approaches, `AdaptiveSoftmax` does not require auxiliary knowledge, is adaptive on a per instance basis, and provides provable guarantees for the true softmax computation directly rather than a proxy.

Our algorithm is inspired by adaptive sampling techniques from the multi-armed bandit literature. Randomized algorithms based on multi-armed bandit algorithms have seen a surge of recent work, due to their ability to provide instance-adaptive guarantees for a variety of problems. This idea was first formalized in the specific case of Monte Carlo Tree Search [26] and later studied in the context of hyper-parameter tuning [31]. Recent work has formalized this approach into the framework of Bandit-Based Monte Carlo Optimization [4], where the computational task is reduced to one of statistical estimation that is solved efficiently with adaptivity. Applications of this framework include finding the medoid of a dataset [5, 7, 41], $k$-nearest neighbor graph construction [30, 34], Monte Carlo permutation-based multiple testing [46], and an adaptive singular value decomposition (SVD) [24]. Most relevant is the recent work of [8], where the authors provide a general framework for adaptive sampling to approximate function evaluation at unknown but estimable points. This work provides general guarantees, but requires a bound on the Lipschitz factor of the function's gradients as input and has potentially poor performance on specific function classes due to its generality.

A sub-problem in our softmax approximation is identifying the index of the largest component; this is equivalent to the Maximum Inner Product Search (MIPS) problem on the preceding matrix-vector product. MIPS is a common problem that has inspired many directions of research [22, 32]. Many of these algorithms focus on specific use cases and place restrictive assumptions on the data (e.g., that all elements of the matrix-vector product are positive), require preprocessing, are not adaptive to the underlying data distribution, or lack PAC guarantees. One large family of MIPS algorithms are based on locality-sensitive hashing (LSH) [19, 2]. In addition to significant preprocessing overhead and practical implementation issues, a shortcoming of these LSH-based approaches is that the maximum dot product is often small compared to the vector norms in high dimensions, which necessitates many hashes and significant storage space (often orders of magnitude more than the data itself). Promising LSH-based algorithms have recently been applied to the problem of softmax computation [1, 15]. These methods focus on intensive preprocessing and work primarily by attaining gains in terms of $n$. In contrast, `AdaptiveSoftmax` subsamples matrix-vector products and obtains gains with respect to $d$. Furthermore, `AdaptiveSoftmax` provides an instance-adaptive algorithm with no required preprocessing and still has PAC guarantees.

## 3  Problem Formulation

In this work, we focus on the problem of identifying the $k$ largest entries of the softmax of an input vector that is the result of a computationally expensive matrix-vector multiplication. Specifically, we analyze the setting where the input vector $\mu$ is the result of a matrix vector product $Ax$, as is common in the final linear layer of neural networks (such scenarios frequently arise in other machine learning problems as well [4]). Our objective is to design an algorithm that can with probability at least $1 - \delta$ estimate the top $k$ values to multiplicative accuracy $\varepsilon$, where $\varepsilon$ and $\delta$ are given input parameters. For clarity of exposition, we focus on the case of $k = 1$, i.e., identifying and estimating the largest component. All our theoretical results, however, easily extend to the setting $k > 1$ (discussed in Section 4.2).

**Notation:** We use $[n]$ to denote the set $\{1, 2, \ldots, n\}$ and $\|\cdot\|$ to denote the vector $\ell_2$ norm, unless otherwise specified. We use $\|\cdot\|_{\Psi_2}$ to denote the Orlicz norm (i.e., the *sub-Gaussianity* parameter or *variance proxy*) of a random variable; this is discussed in greater detail in Appendix A.2 [44]. For matrix $A$ and vector $x$, we denote the resulting product as $\mu = Ax$. Assuming for notational

simplicity that the arms are in sorted order $\mu_1 > \mu_2 \geq \ldots \mu_n$, we define the gaps between the entries of $\mu$ as $\Delta_i = \mu_1 - \mu_i$. We use the convention from the best-arm identification literature that $\Delta_1 = \Delta_2$ and assume that $\Delta_2 > 0$ (this assumption is easily relaxed). Furthermore, we define $\alpha_i = e^{\beta\mu_i}$ and $\gamma_i = e^{\beta\mu_i/2}$, which are proportional to the optimal first (respectively, second) order sampling frequencies; these are discussed further in Section B.2. Finally, we define $\boldsymbol{p}$ as the softmax output, and $i^*$ as its largest entry (assumed to be unique), i.e.,

$$\sigma_\beta(Ax) = \boldsymbol{p}, \quad i^* = \underset{i \in [n]}{\operatorname{argmax}} \, p_i.$$

Our goal is to design an algorithm which efficiently outputs the best index $i^*$ and an estimate its value where, with probability at least $1 - \delta$, the best index is correct and the estimated value is within a factor of $\epsilon$ multiplicative accuracy. Mathematically, defining the algorithm's outputs as $\widehat{i^*} \in [n]$ and $\hat{p}_{i^*} \in [0, 1]$, we define the success events $E_{\text{id}}, E_{\text{est}}$ where the algorithm identifies the largest entry, and where it estimates its value to within multiplicative accuracy $\epsilon$. We define the algorithm as providing $(\varepsilon, \delta)$-PAC guarantees if these events happen simultaneously with probability at least $1 - \delta$, with respect to the randomness of the algorithm.

$$E_{\text{id}} = \left\{ \widehat{i^*} = i^* \right\} \tag{2}$$

$$E_{\text{est}} = \left\{ (1 - \varepsilon)p_{i^*} \leq \hat{p}_{i^*} \leq (1 + \varepsilon)p_{i^*} \right\}. \tag{3}$$

$$\mathbb{P} \left( E_{\text{id}} \cap E_{\text{est}} \right) \geq 1 - \delta \tag{4}$$

Our objective them becomes to design an algorithm that satisfies Equation (4) and minimizes the requisite sample complexity.

## 4 `AdaptiveSoftmax` Algorithm

We now introduce the `AdaptiveSoftmax` Algorithm, which approximates the output of the softmax in Algorithm 1. First, `AdaptiveSoftmax` approximates the softmax normalization constant $Z_\beta$ to a multiplicative accuracy of $\epsilon/4$ via `NormalizationEstimation` (Algorithm 2). Next, `AdaptiveSoftmax` identifies the best arm (or top $k$ arms, depending on the setting) using a standard multi-armed bandit algorithm, `BestArmId` (Algorithm 3). In our setting, "arms" correspond to different rows of $A$ and pulling arm $i$ corresponds to computing a coordinate-wise scalar product $A_{i,j}x_j$ for some coordinate $j$ (we provide a more formal overview of the best-arm identification problem and the associated algorithm in Appendix A). Finally, `AdaptiveSoftmax` estimates the value of the identified best arm (or top $k$ arms) to a multiplicative accuracy of $\epsilon/4$ by sampling each arm a sufficient number of times via `EstimateArm` (Algorithm 10).

We prove $(\varepsilon, \delta)$-PAC guarantees for `AdaptiveSoftmax` by union-bounding over the error probabilities in each step of Algorithm 1. Our results will show that, with probability at least $1 - \delta$, `AdaptiveSoftmax` is able to identify the largest output of the softmax function and estimate its value to multiplicative accuracy $\epsilon$.

---

**Algorithm 1** Adaptive Softmax

1: **Input:** Matrix $A$, vector $x$, temperature $\beta$, error $\epsilon$, failure probability $\delta$, variance proxy $\sigma^2$
2: **Output:** $\hat{p}_{i^*}$ and $\widehat{i^*}$, highest softmax probability and its index
3: # *Estimate denominator of softmax*
4: $\hat{Z} \leftarrow$ `NormalizationEstimation`$(A, x, \beta, \epsilon/4, \delta/3, \sigma^2)$
5: # *Compute index of best arm.*
6: $\widehat{i^*} \leftarrow$ `BestArmId`$(A, x, \delta/3, \sigma^2)$
7: # *Estimate value of best arm*
8: $\hat{\mu}_{i^*} \leftarrow$ `EstimateArm`$(A_{i^*}, x, \epsilon/4, \delta/3)$
9: $\hat{p}_{i^*} = \exp(\beta\hat{\mu}_{i^*})/\hat{Z}$
10: **return** $\hat{p}_{i^*}, \widehat{i^*}$

---

These inputs are typical in the multi-armed bandit setting, but the variance proxy $\sigma^2$ merits additional discussion. In order for our random-sampling-based approach to succeed, a bound on the rate of

---

**Algorithm 2** `NormalizationEstimation`

---

1: **Input:** Matrix $A$, vector $x$, temperature $\beta$, target error $\epsilon$, failure probability $\delta$, variance proxy $\sigma^2$
2: **Output:** $\hat{Z}_\beta$, estimate of the partition function
3: Compute $\hat{\mu}_i$ using $T_0 = 17\beta^2\sigma^2 \log(6n/\delta)$ coordinate samples for each arm
4: $C_i \leftarrow \sqrt{\dfrac{2\sigma^2 \log\left(\frac{6n}{\delta}\right)}{T_0}}$
5: $\hat{\alpha}_i \leftarrow e^{\beta(\hat{\mu}_i - C_i)}$
6: $\hat{\gamma}_i \leftarrow e^{\beta(\hat{\mu}_i - C_i)/2}$
7: Sample each arm $n_i = \min(\tilde{n}_i, d)$ times to recompute the estimates $\hat{\mu}_i$, where

$$\tilde{n}_i = \beta^2\sigma^2 \max\left(17\log\left(\frac{6n}{\delta}\right), \frac{16\sqrt{2}\log\left(\frac{6n}{\delta}\right)\left(\sum_j \hat{\gamma}_j\right)\hat{\gamma}_i}{\epsilon \sum_j \hat{\alpha}_j}, 16\log\left(\frac{12}{\delta}\right)\epsilon^{-2}\frac{\hat{\alpha}_i}{\sum_j \hat{\alpha}_j}\right)$$

8: **return** $\widehat{Z_\beta} = \sum_i e^{\beta\hat{\mu}_i}$

---

concentration of the estimators $\hat{\mu}_i$ is required; the quantity $\sigma^2$ governs the concentration rate, as we discuss in Appendix A. In practice, such a bound holds very generally, for example as long as $A$ and $x$ have bounded entries. For algorithmic simplicity we utilize the following assumption.

**Assumption 1.** *We assume that we are given a variance proxy bound $\sigma^2$ for the sub-Gaussian parameters of the constructed estimators:*

$$\sigma^2 \geq \|A_{iJ}x_J\|_{\Psi_2} \ \forall i, \quad \text{for } J \sim \text{Unif}([n]).$$

We provide theoretical guarantees for `AdaptiveSoftmax` under Assumption 1. Recall that we defined our optimal first and second order sampling frequencies $\alpha_i = e^{\beta\mu_i}$ and $\gamma_i = e^{\beta\mu_i/2}$ (see Appendix B.2). We first show in Proposition 1 that our softmax normalization estimation algorithm (Algorithm 2) obtains the desired guarantees.

**Proposition 1.** *For input $\varepsilon \in (0, 1/2)$, $\delta \in (0, 1)$, and $\sigma$ satisfying Assumption 1, Algorithm 2 will, with probability at least $1 - \delta$, estimate $Z_\beta = \sum_j e^{\beta\mu_j}$ to a multiplicative accuracy of $\epsilon$. On this success event, Algorithm 2 requires at most*

$$C\beta^2\sigma^2\left(n\log\left(\frac{n}{\delta}\right) + \log\left(\frac{n}{\delta}\right)\left(\sum_j \gamma_j\right)^2\left(\epsilon\sum_j \alpha_j\right)^{-1} + \log\left(\frac{1}{\delta}\right)\varepsilon^{-2}\right)$$

*samples for some absolute constant $C$, where non-asymptotic bounds with numerical constants are provided in Appendix B.*

With the sample complexity of Algorithm 2 bounded, the complexity of best arm identification and the cost of estimating the best arm to a target accuracy are readily available from the multi-armed bandit literature. This enables us to state an overall result for `AdaptiveSoftmax` (Algorithm 1) in the following Theorem.

**Theorem 1.** *For input $\varepsilon \in (0, 1/2)$, $\delta \in (0, 1)$, and $\sigma$ satisfying Assumption 1, Algorithm 1 identifies the largest component in $\sigma_\beta(Ax)$ and estimates its value to a multiplicative accuracy of $\epsilon$ with probability at least $1 - \delta$, as in (4). On this success event, the algorithm uses $T$ samples where*

$$T \leq C\sigma^2\left(\beta^2 n\log\left(\frac{n}{\delta}\right) + \sum_{i=1}^{n}\frac{\log\left(\frac{n\log d}{\delta}\right)}{\Delta_i^2} + \frac{\beta^2\log\left(\frac{n}{\delta}\right)\left(\sum_j \gamma_j\right)^2}{\epsilon\sum_j \alpha_j} + \frac{\beta^2\log(1/\delta)}{\varepsilon^2}\right),$$

*for some absolute constant $C$. Tighter bounds with non-asymptotic numerical constants are provided in Appendix B.*

The proofs of these two results are detailed in Appendix B; we provide some intuition and brief sketches of the proofs here.

For Proposition 1, we first show that we can estimate the quantities $\{\alpha_i\}, \{\gamma_i\}$, to constant multiplicative error with high probability. This allows us to construct a sampling scheme based off of the asymptotically optimal sampling frequencies, and guarantee that each arm is sampled at least half of what this asymptotically optimal frequency requires. Then, sampling each arm $i$ enough so that the first order Taylor expansion of $e^{\beta\hat{\mu}_i}$ is sufficiently accurate, we can sample further according to these determined frequencies to guarantee PAC estimation. This is an improved and specialized modification of [8] that exploits the structure of the softmax function to remove the assumption of Lipschitz gradients and yield improved sample complexity (this is discussed further in Appendix B.3).

Next, we utilize a classical best-arm identification algorithm to identify the best arm with high probability, leveraging standard results in Bandit-Based Monte Carlo Optimization [6]. Finally, we sample the identified best arm enough times to estimate its value to multiplicative accuracy $\epsilon/4$ with high probability. By union bounding over these error probabilities, we achieve the desired PAC guarantees.

## 4.1 Interpreting Theoretical Results

We now simplify and further interpret the sample complexity results in Theorem 1. First, note that the $\varepsilon^{-2}$ dependence exhibited by `NormalizationEstimation` (Algorithm 2) is optimal: it is inherent even in estimating the mean of the best arm to accuracy $\varepsilon$. The cost stemming from the second order error, which scales as $\varepsilon^{-1}$, is bounded between $\beta^2\sigma^2\log(n/\delta)\varepsilon^{-1}$ and $n\beta^2\sigma^2\log(n/\delta)\varepsilon^{-1}$, where in the case where one arm is much better than the rest this will match the first term. Concretely, we analyze the setting where the minimum gap is $\Delta$, i.e. $\mu_1 - \mu_i = \Delta_i \geq \Delta$ for all $i$.

**Corollary 1.** *Under the conditions of Theorem 1, when the minimum gap is at least $\Delta$, Algorithm 1 identifies and provides $(\varepsilon, \delta)$-PAC estimation (Equation (4)) of the largest softmax entry, using*

$$T \leq C\left(\beta^2\sigma^2\log\left(\frac{n}{\delta}\right)\left(n + \frac{\varepsilon^{-1}n^2}{n + e^{\beta\Delta}}\right) + \beta^2\sigma^2\varepsilon^{-2}\log(1/\delta) + n\sigma^2\log\left(\frac{n\log d}{\delta}\right)\Delta^{-2}\right)$$

*samples for some universal constant $C$. In the case where the gap is large ($\Delta \geq \frac{2}{\beta}\log n$), $\beta$ is not too small, and $d < e^{e^n}$ (see Equation (47) for precise statement), this can be simplified to*

$$T \leq C\beta^2\sigma^2\left(\log\left(\frac{n}{\delta}\right)\left(n + \varepsilon^{-1}\right) + \varepsilon^{-2}\log(1/\delta)\right).$$

*where all sample complexities are for the $1 - \delta$ success event.*

The proof of this upper bound is in Appendix B.1. Note that this directly implies that when the gap is large (i.e. there is a clear largest output element) and $\varepsilon$ is constant, the sample complexity is nearly linear in $n$ and is upper bounded by $C\beta^2\sigma^2 n\log(n/\delta)$.

## 4.2 Implementation details and extensions

There are many techniques that we can use to extend and improve `AdaptiveSoftmax` in practice. We discuss changes from the written algorithm in detail in Appendix C.

**Randomized Hadamard Transformation:** The variance-proxy bound $\sigma^2$ of the arms plays a large factor in the `AdaptiveSoftmax` algorithm's sample complexity. The underlying sub-Gaussianity parameter of these estimators can be improved using techniques from randomized numerical linear algebra, such as the randomized Hadamard transform [42]. If a small number of entries dramatically increase the variance of the estimator, then the randomized Hadamard transform will make the coordinates more uniform. We provide theoretical guarantees for this approach in Appendix A.3.1.

**Top-$k$ Identification:** Extending our algorithmic results from best arm identification (top 1) to identifying multiple components (top $k$) follows directly from existing multi-armed bandit algorithms. Numerous algorithms have been developed for this setting [21], and variants for computational settings have been developed and studied in [6]. For simplicity and clarity, we focused on the top 1 identification in this paper, but the top $k$ extension readily follows. Furthermore, in numerical experiments we observe estimating the normalization constant $Z_\beta$ dominates the sample complexity, and the increase in cost from identifying the top $k$ arms and estimating their values to multiplicative accuracy $\varepsilon/4$ is minimal.

**Relaxing Assumption of Known sub-Gaussian Parameter** $\sigma^2$**:** Assumptions regarding known arm concentration parameters are common in multi-armed bandit works and simplify theoretical exposition. These results can naturally be extended in several directions. One simple extension is to the setting where we have a separate sub-Gaussianity parameter $\sigma_i^2$ for each arm, i.e., heterogeneous variances. A more practical extension is to the setting where we do not have a bound on the sub-Gaussianity parameter for each arm but know that the arm returns are bounded. In this setting, a common multi-armed bandit approach is to utilize the empirical variance [35]. These approaches are discussed further in [8].

**Improved Estimators** $\hat{\mu}$**:** Naïvely, the `AdaptiveSoftmax` algorithm samples coordinates uniformly at random with replacement from the set of coordinates $\{1, \dots, d\}$ to estimate each $\sum_j A_{ij} x_j$. This procedure can be improved in several ways. For example, we may utilize importance sampling and sample each coordinate with probability $z_j \propto |x_j|$. Furthermore, we can sample coordinates without replacement; this is known to yield tighter confidence intervals than sampling with replacement [9]. We can combine these techniques and compute the effective variance as in [18]. Sampling without replacement can be achieved in a computationally efficient manner via Gumbel sampling [27]. We discuss these details further in Appendix A.3; these details may be of independent interest.

## 5 Experiments

In this section, we demonstrate the empirical advantages of `AdaptiveSoftmax` over the brute-force softmax computation in terms of sample complexity. All of our results are reproducible via a 1-line script, publicly available on GitHub at `github.com/ThrunGroup/adaptiveSoftmax`.

### 5.1 Complexity on Synthetic Data

Crucially, the `AdaptiveSoftmax` algorithm scales sublinearly in $d$. More precisely, Corollary 1 implies that, for fixed $\varepsilon$ and $\delta$, the sample complexity of the `AdaptiveSoftmax` algorithm scales as $O(n \log n)$. We now empirically validate this behavior.

We first run the `AdaptiveSoftmax` algorithm on two synthetic datasets. In each dataset, we generate $A$ and $x$ with $n = 100$ and vary $d$.

In the first synthetic dataset, we set $x$ to be a $d$-dimensional vector of all 1s. We draw each element of $A \overset{\text{i.i.d.}}{\sim} \mathcal{N}(0,1)$ and add the vector of all 1s to the first row of A, thereby planting a signal. In expectation, the first row of $A$ will have inner product $d$ with $x$ whereas all other rows will have inner product 0 with $x$. Furthermore, all arms have expected variance $\sigma_i^2$ that scales with $d$.

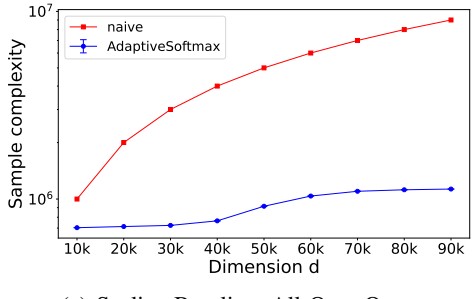 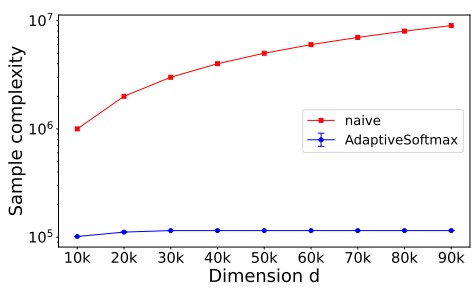

(a) Scaling Baseline: All-Ones Query      (b) Scaling Baseline: Sign Query

Figure 1: Sample complexity of the `AdaptiveSoftmax` algorithm and the brute-force softmax computation on two different synthetic datasets as a function of $d$. Error bars are obtained from 100 random trials. The sample complexity of the `AdaptiveSoftmax` algorithm scales with respect to $d$ for (a) but does not for (b), as expected. The average gains for $\delta = 10\%$ and $\varepsilon = 30\%$ are $3.953\times$ for (a) and $29.039\times$ for (b), increasing with increasing dimension. Confidence intervals are 1std.

In the second synthetic dataset, we draw each element of $A \overset{\text{i.i.d.}}{\sim} \mathcal{N}(0,1)$ and set $x$ to be $|A_{1,:}|$, the entrywise absolute value of the first row of $A$. Here, arms have expected variance $\sigma_i^2 = \Theta(1)$.

Figures 1(a) and 1(b) demonstrates the scaling of the `AdaptiveSoftmax` Algorithm on each of the two datasets. On the first synthetic dataset, the `AdaptiveSoftmax` algorithm scales with $d$ because the variance proxies $\sigma_i^2$ do. On the second synthetic dataset, the `AdaptiveSoftmax` algorithm does not exhibit significant scaling with $d$. On both datasets, the `AdaptiveSoftmax` algorithm significantly outperforms the naïve brute-force computation of the softmax function.

## 5.2 Multinomial Logistic Regression

Multinomial logistic regression (MNL) is a form of multiclass classification in which the final operation performed by the classifier is of the form:

$$P(y = c) = \frac{e^{\beta(w_c^\top h(x))}}{\sum_{c'=1}^{C} e^{\beta(w_{c'}^\top h(x))}} \tag{5}$$

i.e., the probabilities that datapoint $x$ belongs to each class $c$ is given by the softmax applied to the vector $W h(x)$, where $W$ is the matrix containing rows $w_1, \ldots, w_c$ and $h(x)$ is a latent representation of $x$ (i.e., the forward pass of some neural network on $x$).

The multinomial logistic regression is naturally amenable to accelerated softmax computation in Equation (5). In many real-world settings, both the number of classes $C$ and the dimension of the latent representation $h(x)$ (and therefore the dimensionality of each $w_c$) can be very large, motivating the usage of `AdaptiveSoftmax` to identify and estimate the probability of the most likely class. However, the application of `AdaptiveSoftmax` extends far past vanilla MNL. For instance, the final layer of any neural network classifier utilizing softmax can also be viewed as an MNL problem. We now provide several such practical settings for which we demonstrate the benefits of applying the `AdaptiveSoftmax` algorithm.

## 5.3 `AdaptiveSoftmax` Performance on Real Data

We now demonstrate the performance of the `AdaptiveSoftmax` algorithm on several real-world datasets. For each setting, we provide the sample complexity gain relative to the sample complexity of the brute-force, naïve softmax computation sample complexity $nd$. We also provide the success rate of our algorithm in each setting, i.e., the proportion of times the `AdaptiveSoftmax` algorithm correctly identifies the maximum likelihood output (i.e. $\hat{i}^\star = i^\star$) and estimates its probability $p_{i^\star}$ within a multiplicative error of $\varepsilon = 30\%$.

### 5.3.1 Application to CNNs

We consider the application of `AdaptiveSoftmax` to CNN classifiers on two distinct image classification datasets:

1. The MNIST dataset, containing black and white images of handwritten digits as input and ten output classes representing all ten possible digits.
2. The EuroSAT dataset, containing RGB satellite imagery as input and ten output classes, representing possible land types (e.g., river, residential, etc)

On both of these datasets and for distinct architectures, we show that `AdaptiveSoftmax` provides a drastic improvement in sample efficiency.

**MNIST** For the MNIST dataset, we train a shallow CNN from scratch with two convolutional blocks (Conv2d, ReLu, MaxPool, BatchNorm). This model achieves over 99% accuracy on the test set. The matrix $A$ is obtained by extracting the weight matrix of the model's final linear layer. The vector $x$ is extracted as the output of the final hidden layer (the layer before the final linear layer) constructed by passing the MNIST image through the trained model and flattening the result. The dimensionality of $x$ is adjusted by changing the number of output channels of the convolution blocks. The sample complexity of our algorithm is measured by running the algorithm on 1000 different images in test set with same matrix $A$. The empirical error rate $\delta$ is calculated as the fraction of experiments where the adaptive algorithm fails to identify the same class, or fails to estimate the probability to accuracy $\epsilon$, as assigned by exact computation.

**EuroSAT**   We also utilize a larger pre-trained CNN classifier fine-tuned on the EuroSAT dataset, to show that `AdaptiveSoftmax` works with larger more sophisticated CNNs. Specifically, we freeze all convolution blocks of VGG-19 (pretrained on ImageNet) and changed the final output dimension to 10 classes for EuroSAT without freezing the weights. The resulting model achieves 92% accuracy on the test set. As before, the matrix $A$ can be extracted from the weights of the final linear layer and the vector $x$ represents the final hidden layer activations. The empirical error rate $\delta$ is calculated in the same manner as for MNIST.

| Dataset (Model) | $\delta = 10\%$ | $\delta = 5\%$ | $\delta = 1\%$ |
|---|---|---|---|
| EuroSAT (VGG-19) | 5.18x (80.62%) | 5.16x (83.00%) | 4.54x (98.37%) |
| MNIST (Shallow CNN) | 8.95x (92.25%) | 8.81x (93.75%) | 8.13x (99.38%) |

Table 1: Performance improvement and success rate afforded by `AdaptiveSoftmax` for multinomial logistic regression on two different real-world datasets. We used a total of $q = 800$ test queries to measure success rate.

### 5.3.2   Application to LLMs

We also apply the `AdaptiveSoftmax` algorithm to LLMs using HuggingFace's `AutoModelForCausalLM` module for the task-generation task [45]. The matrix $A$ is the `lm-head` layer for each model, and the queries $x$ are the final hidden states of the model that is extracted by running a forward pass of the model on the given dataset with a window moving at a certain stride. The context window and stride is modified to generate a desired number of queries.

| Dataset (Model) | $\delta = 10\%$ | $\delta = 5\%$ | $\delta = 1\%$ |
|---|---|---|---|
| Wikitext (GPT-2) | 8.25x (88.94%) | 7.80x (93.54%) | 6.67x (98.26%) |
| Wikitext (Llama3-7B) | 14.68x (91.44%) | 11.43x (94.04%) | 6.88x (99.38%) |
| Wikitext (Mistral7B) | 32.65x (89.08%) | 26.37x (91.20%) | 17.71x (97.77%) |
| Penn Treebank (GPT-2) | 8.10x (81.68%) | 7.50x (90.73%) | 6.66x (96.79%) |
| Penn Treebank (Llama3-7B) | 19.18x (87.82%) | 16.57x (91.60%) | 10.72x (97.81%) |

Table 2: Performance improvement and success rate afforded by `AdaptiveSoftmax` for LLM inference (improvement for final softmax layer). Experiment details in Section 5.3.2. We used $q = 1000$ unseen test queries to measure $\delta$-accuracy.

Our matrix $A$ is the extracted lm-head from HuggingFace's AutoModelForCausalLM for the four models: GPT-2 ($n = 50257, d = 768$), Llama3-7B ($n = 128256, d = 4096$), Mistral7B ($n = 32000, d = 4096$), and Gemma7B ($n = 256000, d = 3072$). Our task is task-generation, and we generate our queries $x$ by using two datasets (Wikitext and Penn Treebank) with a sliding window of certain stride. Stride and context window is set to get $q = 1000$ number of queries. Constants and confidence intervals given by theory are empirically quite loose, so we tuned algorithm parameters (constant coefficients for stage length and confidence interval width) on initial training data, described in Appendix C. An aggressive tuning strategy was undertaken in order to demonstrate the potential gains in sample complexity provided by `AdaptiveSoftmax` . Specifically, constant multiples were applied to variance estimate within Algorithm 3 and Algorithm 2. Due to the limited sample set, this approach occasionally overoptimized the constants on training data, yielding lower success rates than targeted. However, from the results, it is clear that this target parameter still provides users sufficient control over the tradeoff between true success rate and sample complexity.

## 6   Discussion, Limitations, and Future Work

In this work, we proposed a theoretically novel and practically efficient algorithm for approximating the softmax function. We provided theoretical guarantees on the accuracy of our approximation and demonstrated that, with fewer samples than exact computation, we can approximate the softmax function to the desired accuracy. We further demonstrated the viability of our proposed algorithm in two real-world settings, multinomial logistic regression and LLM inference.

A potential limitation of our proposed algorithm is that it is most beneficial when the inner dimension of the matrix vector product is high dimensional; its benefits over exact computation are more modest

when the inner dimension is small. In particular, the exact computation of the matrix-vector product preceding a softmax operation is usually performed efficiently using BLAS (Basic Linear Algebra Subroutines, which are highly optimized). Adaptivity at its core is inherently sequential, whereas BLAS operations take advantage of batch computation. In this work we proposed minimally adaptive algorithms, with only a logarithmic number of rounds of adaptivity, but there are important directions of future work to realize these theoretical gains in practice.

**Limitations:** Theoretical sample complexity bounds are useful for understanding the fundamental properties of an algorithm, but in practice, wall-clock time is often the metric of interest. Many of the steps in our algorithm can be batched and made BLAS efficient, yielding comparable wall clock times to brute force computation. However, in general adaptivity is the opposite of batching, as can be seen when we modify our algorithm to adapt to arm specific variances. In this case, we must sample each arm individually, as the number of samples required for each arm is different. This is a trade-off between adaptivity and wall-clock time, and in practice, the choice of which to prioritize depends on the specific application (energy efficiency, computational resources, etc.). There are also possible theoretical analyses, where we can e.g. create batches of arms with similar empirical variance and sample all arms within a batch together, leading to a trade-off between adaptivity and batched computational efficiency. Additionally, in large language models, the final softmax layer is often not a computationally significant step, so while such a method may greatly accelerate multinomial logistic regression, more work may be required to have this accelerate LLMs.

Given the ubiquity of the softmax function in today's machine learning workflows, we hope that our algorithm will help pave the way for an optimized adaptive softmax that can accelerate a wide class of machine learning models. An interesting direction of future work is trying to combine this multi-armed bandit approach with LSH [15] to obtain (for the attention case) subquadratic complexity in $n$, and sublinear complexity in $d$. The adaptive method we propose for estimating the normalizing constant of the softmax function is of independent interest, and holds potential for applications in kernel density estimation and other machine learning tasks.

# Acknowledgements

Mert Pilanci was supported in part by the National Science Foundation (NSF) under Grant DMS-2134248; in part by the NSF CAREER Award under Grant CCF-2236829; in part by the U.S. Army Research Office Early Career Award under Grant W911NF-21-1-0242; and in part by the Office of Naval Research under Grant N00014-24-1-2164. Tavor Baharav was supported by funding from the Eric and Wendy Schmidt Center at the Broad Institute of MIT and Harvard.

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

# A Bandit preliminaries

To make this work accessible to a broad audience, we provide a self contained introduction to the multi-armed bandit setting.

## A.1 Best arm identification

We consider a stochastic multi-armed bandit problem [13, 40, 29] with $n$ arms (distributions), where each arm $i$ has an unknown mean reward $\mu_i$. At each time step $t$, the algorithm selects an arm $I_t \in [n]$ and receives a reward $X_{I_t,t}$ drawn from the distribution of arm $I_t$. Early work in the multi-armed bandit literature focused on the regret minimization setting, where the goal is to maximize the cumulative reward (sum of arm pulls observed so far), motivated by applications such as online advertising and gambling [28]. Recent work has seen increased interest in the best-arm identification setting, where the goal is to identify the arm with the highest mean reward with high probability, motivated by applications such as clinical trials. Significant research has been devoted to obtaining optimal logarithmic factors, but for the sake of clarity we highlight here a simpler and empirically well performing algorithm, `multi-round` $\epsilon$-`arm` from [25].

---

**Algorithm 3** `BestArmId` (modification of best-arm identification from [25])

---

**Input:** $n$ arms, error probability $\delta$, variance proxy $\sigma^2$
**Output:** Best arm $i^\star$ with probability at least $1 - \delta$
$S_0 \leftarrow [n]$
$r \leftarrow 0$
$t_0 \leftarrow 0$
**while** $|S_r| > 1$ **do**
    $r \leftarrow r + 1$
    $\epsilon_r \leftarrow 2^{-r}$
    $t_r \leftarrow \lceil 8\sigma^2 \epsilon_r^{-2} \log(4nr^2/\delta) \rceil$
    **for all** arms $i \in S_{r-1}$ **do**
        Pull arm $i$ $t_r - t_{r-1}$ times and observe rewards $X_{i,t_{r-1}+1}, \ldots, X_{i,t_r}$
        $\hat{\mu}_{i,r} \leftarrow \frac{1}{t_r} \sum_{s=1}^{t_r} X_{i,s}$          $\triangleright$ Update mean estimates
        $C_{i,r} \leftarrow \sqrt{2\sigma^2 \log(4nr^2/\delta)/t_r}$      $\triangleright$ Compute confidence interval width
    **end for**
    Set $S_r \leftarrow \{i \in S_{r-1} : \hat{\mu}_{i,r} + C_{i,r} \geq \max_{j \in S_{r-1}} \hat{\mu}_{j,r} - C_{j,r}\}$      $\triangleright$ Filter "bad" arms
**end while**
**return** $i^*$, the only element in $S_r$      $\triangleright$ We assume $i^*$ is unique (this assumption is easily relaxed)

---

The algorithm proceeds in rounds, maintaining a set of arms $S_r$ that are still in contention for being the best arm. At each round $r$ the algorithm pulls each surviving arm such that we can construct a high probability confidence interval of width $\epsilon_r/2$ around the mean of each arm. Then, any arms whose empirical mean plus confidence interval is less than the maximum empirical mean minus confidence interval are eliminated. If this is the case, then that arms mean is with high probability less than the maximum mean, and so it is eliminated from contention. This preserves the best arm with high probability, and the algorithm terminates when only one arm remains (this best arm).

## A.2 Sub-gaussian random variables

Following the exposition of [44], for a strictly increasing convex function $\psi : \mathbb{R}_+ \rightarrow \mathbb{R}_+$ where $\psi(0) = 0$, the $\psi$-Orlicz norm of a random variable $X$ is defined as:

$$\|X\|_\psi \triangleq \inf \left\{ t > 0 | \mathbb{E} \left[ \psi \left( t^{-1}|X|| \right) \right] \leq 1 \right\}.$$

$\|X\|_\psi$ is infinite if the expectation $\mathbb{E} \left[ \psi \left( t^{-1}|X|| \right) \right]$ does not exist for any finite $t$. The sub-Gaussian parameter of a random variable is defined as the $\psi_2$-Orlicz norm, where $\psi_2(u) = e^{u^2} - 1$.

Standard results (Hoeffding's lemma) provide that for a random variable $X$ such that $a \leq X \leq b$ almost surely, that $\|X\|_{\psi_2} \leq \frac{(b-a)^2}{4}$.

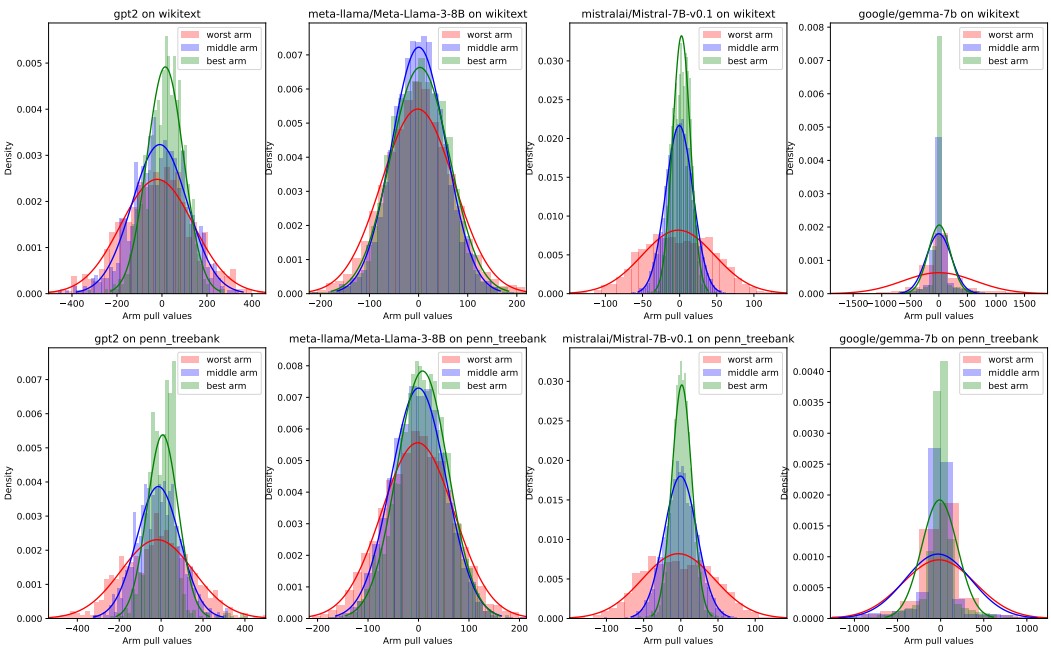

Figure 2: Distribution of arm pulls for the best, middle, and worst arms for a random query. The Gaussian fit plotted for each arm is computed using the empirical variance of the given arm, and can be seen to closely match the empirical distribution, indicating that the arm pull distributions are well approximated by a Gaussian. This merits the assumption of sub-Gaussianity. Arm pulls represent importance weighted samples based on the magnitude of the query vector, so for an arm $i$ and sample $j$, the value is $A_{i,j} * \text{sgn}(x_j) * d$. $\sigma$ is computed across all samples for an arm, and windows are truncated at the lowest and highest ends of the $3\sigma$ ranges across arms for viewing clarity.

Hoeffding's inequality provides a useful concentration bound, where for $X$ with $\|X\|_{\psi_2} \leq \sigma^2$,

$$\mathbb{P}\left(|X - \mathbb{E}[X]| \geq t\right) \leq \exp\left(-\frac{t^2}{2\sigma^2}\right).$$

### A.2.1 Sub-gaussianity in practice

The assumption of sub-Gaussianity is the only assumption that we make in this paper. It is one of the weakest assumptions possible (does not assume that the arms are Bernoulli or Gaussian), and is a common assumption in the multi-armed bandit and adaptive computation literature [4]. Unfortunately, without such an assumption, no nontrivial results are possible; consider the case where we do not have preprocessing access to $A$, the vector $x$ is all 1s, and $A$ is all 1s except for a randomly selected entry which has value 2. In this case, any algorithm for PAC computation of softmax$(Ax)$ with $\delta = 1 - 1/n$ (even just identification of the largest entry of $Ax$) requires $\Omega(nd)$ samples. More practically though, these vectors are the result of a machine learning pipeline, and not of adversarial construction. As shown by our simulations, this worst case scenario never occurs in practice, and arm pulls are generally well approximated by a Gaussian (see Figure 2). Additionally, note that for any fixed problem instance, all arm pulls are bounded, and are thus sub-Gaussian.

### A.3 Improved estimators

To provide theoretical guarantees in multi-armed bandit problems, stringent assumptions are often required, e.g. Assumption 1. This is so that we can provide high probability guarantees on the concentration of the estimator constructed as the empirical mean of the observed samples. Often, these assumptions are phrased as either that random variables corresponding to arm pulls are bounded in $[0, 1]$ a.s., or that they are $\sigma^2$ sub-Gaussian, with a known bound on $\sigma^2$. Such analyses have been generalized to bounded random variables with a known bound, where the algorithm is able to

adapt to the unknown variance [33, 35]. In this work, as is often done to make multi-armed bandit algorithms more performant [4], we instead directly use the empirical variance of an estimator as its sub-Gaussian parameter. In the Gaussian case, a random variable's sub-Gaussian parameter $\sigma^2$ matches its variance: as our arm pulls are constructed as the sum of many $(d)$ terms, which can be thought of as weakly dependent, our arm pulls can be thought of as Gaussian random variables with variance $\sigma^2$. Since in practice, we do not have a good bound on the magnitude of these arm pulls, we directly use Hoeffding's concentration inequalities [44] with the empirical variance $\hat{\sigma}^2$, as opposed to an empirical Bernstein type concentration inequality [35].

### A.3.1 Randomized Hadamard Transform

We discuss applying a randomized Hadamard transform to reduce the sub-Gaussian parameter of our estimators. Define the rotation matrix $R = \frac{1}{\sqrt{d}}DH$, where $D$ is a diagonal matrix with diagonal entries equiprobably $\pm 1$. $H$ is a Hadamard matrix ($d$ must be a power of 2, 0 padded if necessary). Then, we have that applying the transform $R$ to $A$ and $R^\top$ to $x$ yields arms with better sub-Gaussian parameters. Concretely, define $Z = AR$ and $y = R^\top x$. Analyzing $y$ first, we have that no entry of $y$ is too large, as:

$$\mathbb{P}(\|y\|_\infty \geq t) \leq \sum_{i=1}^{n} \mathbb{P}\left(\left|R_i^\top x\right| \geq t\right) \tag{6}$$

$$= n\mathbb{P}\left(\sum_{j=1}^{d} \xi_j d^{-1/2} x_j \geq t\right) \tag{7}$$

$$\leq 2n \exp\left(-\frac{2t^2}{4\sum_{j=1}^{d} \frac{1}{d}x_j^2}\right) \tag{8}$$

$$= 2n \exp\left(-\frac{t^2}{\frac{2}{d}\|x\|_2^2}\right). \tag{9}$$

The first inequality is a union bound over the $n$ points and plugs in for $y = R^\top x$. The second equality uses the fact that the Hadamard matrix multiplied by the random diagonal $\pm 1$ matrix $D$ makes $R_i$ i.i.d. $\pm 1$, and we use $\xi_j$ to denote these Rademacher i.i.d. $\pm 1$ random variables. Next we use Hoeffding's inequality. Finally, we simplify.

Thus, with probability at least $1 - \delta$, $\|y\|_\infty < \|x\|_2 \sqrt{\frac{2\log(2n/\delta)}{d}}$.

A similar argument can be made for $Z = AR$, showing that each entry in the $i$-th row is upper bounded by $\|A_i\|_2 \sqrt{\frac{2\log(2nd/\delta)}{d}}$, holding simultaneously for all $i, j$.

Since $Ax = Zy$, we can use bandits to approximate $Zy$ instead of $Ax$. With the above analysis, we have a bound on the maximum entry of $Z_{iJ}y_J$, giving us a bound on the sub-Gaussian parameter of each arm with probability $\geq 1 - \delta$.

$$\max_{ij} |Z_{ij}y_j| = \max_i \|A_i\|_2 \|x\|_2 \frac{2\log(4nd/\delta)}{d} \tag{10}$$

Whereas before:
$$\max_{ij} |A_{ij}x_j| = \max_i \|A_i\|_\infty \|x\|_\infty \tag{11}$$

In practice, we did not observe that this transformation yielded much improved variance, as opposed to simply importance sampling. Thus, we do not utilize this in our main algorithm.

### A.3.2 Importance sampling

Instead of naively sampling each coordinate uniformly, we can construct improved estimators using importance sampling. Concretely, consider the unbiased estimator $Z$ where $\mathbb{P}(Z = \frac{1}{p_j}A_jx_j) = p_j$

for some probability distribution $\{p_j\}$ where $p_j > 0$ for all $j$ and $\sum_j p_j = 1$. Now, we are left with the design choice of how to construct $\{p_j\}$; naively, this is $1/d$. Unpacking the variance of our estimator, we see:

$$\text{Var}(Z) = \sum_j \frac{1}{p_j} \left(A_j x_j\right)^2 - \mu^2. \tag{12}$$

Consider the case where $A_j \overset{\text{i.i.d.}}{\sim} Q$ for some distribution $Q$ with $\mathbb{E}[Q^2] = \lambda^2$ (an empirically not unreasonable assumption), where $\{x_j\}$ are fixed constants. We assume $x$ is known, but $A$ isn't. Then, we simplify the expected variance with respect to this randomness in $A$ as

$$\mathbb{E}\left[\text{Var}(Z)\right] = \mathbb{E}\left[\sum_j \frac{1}{p_j} \left(A_j x_j\right)^2\right] - \mu^2$$

$$= \lambda^2 \sum_j \frac{x_j^2}{p_j} - \mu^2 \tag{13}$$

Simplifying this for $Z_{\text{naive}}$ where $p_j = \frac{1}{d}$, and for $Z_{\text{opt}}$ where $p_j \propto |x_j|$, we have

$$\mathbb{E}[\text{Var}(Z_{\text{naive}})] = d\|x\|_2^2 - \mu^2 \tag{14}$$
$$\mathbb{E}[\text{Var}(Z_{\text{opt}})] = \|x\|_1^2 - \mu^2 \tag{15}$$

In the case where we are using the same matrix $A$ over many vectors $x$ (as is often case e.g. in LLM inference, or multinomial logistic regression), we can make our leverage scores a function of precomputed quantities based off of $A$, not just $x$. In this case, it makes sense to consider forms of leverage sampling; in this work we consider taking $p_j \propto |x_j| |\sum_i |A_{ij}|$.

### A.3.3 Sampling without replacement using Gumbel trick

Weighted sampling with replacement becomes wasteful in larger sample size regimes, for which the same high-weight elements are sampled repeatedly. It therefore becomes desirable to remove sampled elements from consideration after they are sampled, re-weighting the remaining elements accordingly. We could naively repeat this iterative process of sampling, removing, and re-weighting our elements until we ended up with a sample of the desired size, say $k$. However, this process is sequential and quite slow. Fortunately, as noted in [43], sampling with replacement according to a set of weights is equivalent to perturbing the logits $\lambda_{1\ldots n}$ of our desired sample weights with draws from the i.i.d standard Gumbel distribution and taking the elements with the top-$k$ perturbed logits as our sample, as detailed in Algorithm 6 and 7. This process is easily batched and is much faster as a result.

Further, taking the $(k+1)$-th perturbed logit as an empirical threshold $\tau$, the inclusion of an element $j$ in our sample is solely dependent on whether or not its perturbed logit value exceeded this threshold. This derivation is detailed in [27], and gives us the following expression (Gumbel CDF) for the inclusion probability of element $j$ in a set $S$ of size $k$ drawn without replacement according to weights $w$:

$$\pi_j = \mathbb{P}\left(j \in S\right) \tag{16}$$
$$\hat{\pi}_j = 1 - \exp(-\exp(\lambda_j - \tau)) \tag{17}$$
$$\mathbb{E}\left[\frac{\mathbb{1}_{\{j \in S\}}}{\hat{\pi}_j}\right] = \frac{1}{\pi_j} \tag{18}$$

These empirical estimates of the marginal probabilities of selection for each column allow us to generate a sequence of estimators for each arms' mean, with improved variance discussed in the next section.

### A.3.4 Variance estimation for Gumbel Samples

The Gumbel sampling trick used in A.3.3 with the fixed empirical threshold also provides us a different lens on our sampling process. Namely, we can compute these closed form inclusion probabilities $\pi$,

and by setting this empirical threshold, we may treat the inclusion of separate elements as independent. Given these observations (sampled $S$), an unbiased estimator for the variance of $\hat{\mu}_i$, constructed as the importance sampling weighted mean of the observations, can be computed from [18] as:

$$S \sim \text{sample } k \text{ elements without replacement from } [d] \text{ according to } \pi \tag{19}$$

$$\hat{\mu}_i = \frac{1}{|S|} \sum_{j \in S} \frac{1}{\hat{\pi}_j} A_{ij} x_j \tag{20}$$

$$\mathbb{E}\left[\hat{\mu}_i\right] = \mu_i \tag{21}$$

$$\text{Var}(\hat{\mu}_i; \tau) = \sum_{j \in S} (A_{i,j} x_j)^2 \left(\frac{1 - \pi_j}{\pi_j^2}\right) + \sum_{j \neq k}^{n} (A_{i,j} A_{i,k} x_j x_k) \left(\frac{\pi_{j,k} - \pi_j \pi_k}{\pi_{j,k} \pi_j \pi_k}\right) \tag{22}$$

$$\widehat{\text{Var}}(\hat{\mu}_i) = \sum_{j \in S} (A_{i,j} x_j)^2 \left(\frac{1 - \hat{\pi}_j}{\hat{\pi}_j^2}\right) \tag{23}$$

Following the analysis of [18] and assuming that our threshold $\tau$ is fixed, we may conclude that the estimate of the variance of $\hat{\mu}_i$, the quantity $\text{Var}(\hat{\mu}_i; \tau)$, is an unbiased estimate of the true variance $\text{Var}(\hat{\mu}_i)$. Further, in all datasets we analyzed, the entries of $A$ were generally symmetric, zero-mean, and not correlated with the corresponding entries of $x$, as seen in Figure 3(a). Since $\tau$ and the values of $\pi$ are selected solely based on the values of $x$, these two further assumptions make the second order term $(A_{i,j} A_{i,k} x_j x_k) \left(\frac{\pi_{j,k} - \pi_j \pi_k}{\pi_{j,k} \pi_j \pi_k}\right)$ zero in expectation.

Thus, for simplicity, we discard this latter summation and treat $\widehat{\text{Var}}(\hat{\mu}_i)$ as an unbiased estimate of the variance throughout our implementation, which can be computed in linear instead of quadratic time (updated in constant vs linear time). In practice, as observed in Figure 4, this variance estimator $\widehat{\text{Var}}(\hat{\mu}_i)$ (solid green line) provides a far better estimate of the Gumbel sampler's true variance than other methods.

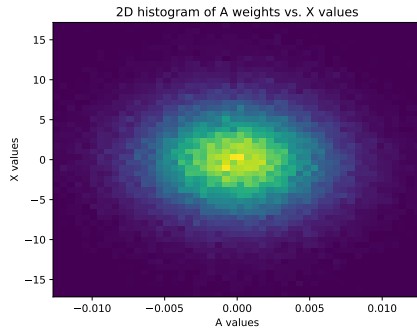

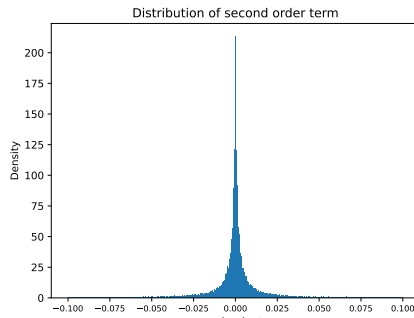

(a) Sampled entries of $A$ and $x$

(b) Sampled values of the second order term

Figure 3: (a) Sampled entries of $A$ and the corresponding entries of $x$ for Mistral on the Wikitext dataset. The values of $A$ are symmetrical about $0$ and not correlated with $x$. (b) Sampled values of the second order term $(A_{i,j} A_{i,k} x_j x_k) \left(\frac{\pi_{j,k} - \pi_j \pi_k}{\pi_{j,k} \pi_j \pi_k}\right)$.

# B    Proofs

We begin by proving a standard best arm identification result, for a slightly modified version of the round-based algorithm from [16].

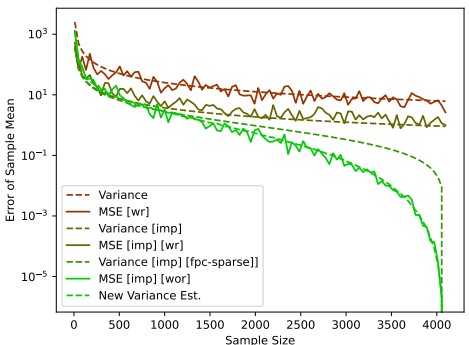

Figure 4: Variance estimates vs. empirical mean squared error. This demonstrates the dramatic improvement afforded by improved estimators and tight confidence intervals. [imp] indicates importance sampling, [wr] with replacement, [wor] without replacement, [fpc-sparse] includes the finite population correction factor.

**Lemma 1** (Best-arm identification). *With probability at least $1 - \delta$, Algorithm `BestArmId` identifies the top softmax value correctly with probability using a number of observations at most*

$$\sum_{i=1}^{n} \min \left( \frac{32\sigma^2 \ln \left( \frac{4n}{\delta} \log_2^2 \left( 4/\Delta_i \right) \right)}{\Delta_i^2}, d \right).$$

*Proof.* Following the proof from [16], since an arm's mean is exactly computed after $d$ pulls, we have that the best arm will be identified with probability at least $1 - \delta$ requiring for arm $i$ a number of pulls at most

$$n_i \leq \min \left( \frac{32\sigma^2 \ln \left( \frac{4n}{\delta} \log_2^2 \left( 4/\Delta_i \right) \right)}{\Delta_i^2}, d \right). \tag{24}$$

Summing over arms yields the desired result, noting that the best arm is pulled at most as many times as it takes to eliminate every other arm (i.e. the second best arm, $\Delta_1 = \Delta_2$).

Note that if $\Delta_i \leq \frac{C}{\sqrt{d}}$, then the second term $(d)$ will be selected, for a sufficiently small absolute constant $C$. Thus,

$$n_i \leq \frac{C\sigma^2 \log \left( \frac{n}{\delta} \log(d) \right)}{\Delta_i^2} \tag{25}$$

Hence, the total sample complexity on this success event is at most

$$
\begin{aligned}
T &= \sum_{i=1}^{n} n_i \\
&\leq \sum_{i=1}^{n} \min \left( \frac{32\sigma^2 \ln \left( \frac{4n}{\delta} \log_2^2 \left( 4/\Delta_i \right) \right)}{\Delta_i^2}, d \right) \\
&\leq C \sum_{i=1}^{n} \min \left( \frac{\sigma^2 \log \left( \frac{n}{\delta} \log(d) \right)}{\Delta_i^2}, d \right)
\end{aligned}
$$

$\square$

We additionally require a lemma for estimating the mean of the best-arm in a PAC sense.

**Lemma 2** (Exponential best arm estimation). *Sampling an arm using*

$$T = \frac{32\sigma^2 \beta^2 \log(2/\delta)}{\epsilon^2}$$

*samples guarantees that $e^{\beta \hat{\mu}_k}$ estimates $e^{\beta \max_i \mu_i}$ to multiplicative accuracy $\epsilon$, with probability at least $1 - \delta$.*

*Proof.* We estimate the mean of arm $k$ after $T$ draws using the plug-in estimator $\hat{\mu}_k$. For simplicity, assume $\beta = 1$, where in the end we scale the sample complexity by $\beta^2$. Sub-Gaussian concentration provides that with probability at least $1 - \delta$,

$$|\hat{\mu}_k - \mu_k| \leq \sqrt{2\sigma^2 \log(2/\delta)/T} = \epsilon/4$$

which in turn implies

$$\log(1 - \epsilon) \leq -\epsilon/4 \leq \hat{\mu}_k - \mu_k \leq \epsilon/4 \leq \log(1 + \epsilon)$$

for $0 < \epsilon < 1$. Then, exponentiating both sides yields

$$1 - \epsilon \leq e^{\hat{\mu}_k - \mu_k} \leq 1 + \epsilon \iff (1 - \epsilon)e^{\mu_k} \leq e^{\hat{\mu}_k} \leq (1 + \epsilon)e^{\mu_k}.$$

Scaling the number of samples by $\beta^2$ yields the desired result. $\qquad\square$

With these two steps in place, we are now ready to tackle the novel technical challenge of this work; estimating the normalization constant of the softmax. We denote the softmax normalization constant as $f(\boldsymbol{\mu}) = \sigma_\beta(\boldsymbol{\mu}) = \sum_i e^{\beta\mu_i}$.

**Proposition 2** (Softmax normalization estimation: restatement of Proposition 1). *Under Assumption 1, Algorithm 2 will, with probability at least $1 - \delta$, estimate $f_\beta(\boldsymbol{\mu}) = \sum_j e^{\beta\mu_j}$ to a multiplicative accuracy of $\epsilon$, using a number of samples at most*

$$T = 2nT_0 + T_1$$

$$= 34\beta^2\sigma^2 \log(6n/\delta)n + \frac{91\sigma^2\beta^2 \log(6n/\delta)\left(\sum_{i=1}^n \gamma_i\right)^2}{\epsilon \sum_i \alpha_i}$$

$$+ \frac{16\beta^2\sigma^2 \log(12/\delta)}{\epsilon^2}$$

*Proof.* To prove Proposition 1, we want to show that with high probability, we can upper and lower bound our plugin estimator as

$$(1 - \epsilon)f(\mu) \leq f(\hat{\mu}) \leq (1 + \varepsilon)f(\mu),$$

giving us our desired multiplicative error bound. We construct several success events that collectively guarantee our bound holds, and that occur with high probability.

- $E_1$ is the event where our estimated optimal sampling frequencies are not too far from the unknown optimal frequencies, i.e.

$$\hat{\alpha}_i \geq \alpha_i/2, \qquad \beta C_i < 1, \quad i = 1, \ldots, n.$$

On this event, we will sample arms sufficiently in the second round.

- $E_2$ is the event where all estimators $\hat{\mu}_i$ are within their 2-sided confidence intervals in stage 2, i.e.

$$|\hat{\mu}_i - \mu_i| \leq \sqrt{2\sigma^2 \log(12n/\delta)/n_i}, \quad i = 1, \ldots, n.$$

On this event, we can bound the error in the exponentiated estimator.

- $E_3$ is the event where the first and second order errors are small, i.e.

$$-\frac{\epsilon}{2}f(\mu) \leq \sum_i e^{\beta\mu_i}\beta(\mu_i - \hat{\mu}_i) \leq \frac{\epsilon}{2}f(\mu)$$

and

$$\sum_i e^{\beta\mu_i}\beta^2(\mu_i - \hat{\mu}_i)^2 \leq \frac{\epsilon}{2}f(\mu).$$

These two terms arise from bounding $f(\hat{\mu})$.

We now show that if $E_1$ and $E_2$ and $E_3$ all occur, then our desired bound holds.

Lemma 3 shows that when $E_2$ holds,

$$f(\hat{\mu}) \leq \sum_i e^{\beta \mu_i} \left(1 + \beta(\mu_i - \hat{\mu}_i) + \beta^2(\mu_i - \hat{\mu}_i)^2\right)$$

$$= f(\mu) + \underbrace{\sum_i e^{\beta \mu_i} \beta(\mu_i - \hat{\mu}_i)}_{(1)} + \underbrace{\sum_i e^{\beta \mu_i} \beta^2 (\mu_i - \hat{\mu}_i)^2}_{(2)}.$$

Note that if $E_3$ holds as well, the expression is upper bounded by $(1 + \epsilon)f(\mu)$, since each of (1) and (2) is bounded above by $(\epsilon/2)f(\mu)$.

All that remains is to show the lower bound also holds. We use the global inequality $1 + x \leq e^x$ to lower bound

$$
\begin{aligned}
f(\hat{\mu}) &= \sum_i e^{\beta \hat{\mu}_i} \\
&= \sum_i e^{\beta \mu_i + \beta(\hat{\mu}_i - \mu_i)} \\
&\geq f(\mu) + \underbrace{\sum_i e^{\beta \mu_i} \beta(\hat{\mu}_i - \mu_i)}_{(1)}.
\end{aligned}
$$

Again, under $E_2$, (1) is lower bounded by $-\epsilon/2 f(\mu)$, which implies our desired lower bound $(1 - \epsilon)f(\mu) \leq f(\hat{\mu})$. Thus, all three events holding guarantees our desired bound holds.

Now that we know our bound holds on the joint success event, all that remains is to show it holds with sufficiently high probability. To do so, we invoke our lemmas which characterize the probability of each event.

The probability of $E_1, E_2$, and $E_3$ all holding is

$$\mathbb{P}(E_1 E_2 E_3) = \mathbb{P}(E_1)\mathbb{P}(E_2|E_1)P(E_3|E_2 E_1).$$

Lemma 3 says that if each arm is sampled $T_0$ times, $P(E_1) \geq 1 - \delta/3$. Lemma 4 says $P(E_2|E_1) \geq 1 - \delta/3$. Lemma 5 and Lemma 6 together show $P(E_3|E_2 E_1) \geq 1 - \delta/3$. Thus, taken together we have

$$
\begin{aligned}
\mathbb{P}(E_1 E_2 E_3) &= \mathbb{P}(E_1)\mathbb{P}(E_2|E_1)P(E_3|E_2 E_1) \\
&\geq (1 - \delta/3)^3 \\
&\geq 1 - \delta,
\end{aligned}
$$

as desired, and so we are done. $\qquad \square$

The lemmas and their proofs follow.

**Lemma 3.** *For $n_i = T_0 = 17\beta^2\sigma^2 \log(\frac{6n}{\delta})$ for all $i$, we have that the event $E_1$,*

$$\hat{\alpha}_i \geq \alpha_i/2, \quad \beta C_i < 1, \qquad i = 1, \ldots, n$$

*occurs with probability at least $1 - \delta/3$.*

*Proof.* By a standard Chernoff bound, with $\sigma^2$ a bound on the sub-Gaussian parameter of all arms, we have that with probability at least $1 - \delta/3$ that for all $i = 1, \ldots, n$,

$$\hat{\alpha}_i = \frac{e^{\beta(\hat{\mu}_i - C_i)}}{\sum_j e^{\beta(\hat{\mu}_j - C_j)}} \geq \frac{e^{\beta(\mu_i - 2C_i)}}{\sum_j e^{\beta \mu_j}} \geq \frac{1}{2}\alpha_i,$$

where $\alpha_i = \frac{e^{\beta \mu_i}}{\sum_j e^{\beta \mu_j}}$, and $C_i = \sqrt{\frac{2\sigma^2 \log(6/n\delta)}{T_0}}$ is the Chernoff confidence interval width constructed such that $C_i < \log(2)/2\beta$ and so $\beta C_i < 1$. To simplify constants, we use that $8/\ln^2(2) < 17$. $\quad \square$

**Lemma 4.** *Sampling as $n_i \geq T_0$ guarantees that conditioned on $E_1$, $E_2$ occurs with probability at least $1 - \delta/3$ and that on $E_2$, $f(\hat{\mu}) \leq f(\mu) + \sum_i \beta(\mu_i + \hat{\mu}_i) + \sum_i \beta^2(\mu_i - \hat{\mu}_i)^2$.*

*Proof.* Suppose we sample each arm $n_i$ times. Note that by a Chernoff bound on each arm,

$$|\hat{\mu}_i - \mu_i| \leq \sqrt{2\sigma^2 \log(6n/\delta)/n_i}$$

holds on each arm independently with probability at least $1 - \delta/3n$, so all arms are within the two-sided bound with probability at least $1 - \delta/3$.

We upper bound the plugin estimator $f(\hat{\mu})$.

$$f(\hat{\mu}) = \sum_i e^{\beta \hat{\mu}_i} \tag{26}$$

$$= \sum_i e^{\beta \mu_i + \beta(\mu_i - \hat{\mu}_i)} \tag{27}$$

$$\leq \sum_i e^{\beta \mu_i} \left(1 + \beta(\mu_i - \hat{\mu}_i) + \beta^2(\mu_i - \hat{\mu}_i)^2\right) \tag{28}$$

where in (28) we use the upper bound $e^x \leq 1 + x + x^2$ for $x \leq 1.79$ on the event $E_2$, since on $E_2$, $(\mu_i - \hat{\mu}_i) \leq 1/\beta$. This is because $n_i \geq T_0$, and $T_0$ samples already guarantees this. $\square$

**Lemma 5** (First-order error concentration). *On the event $E_2 \cap E_1$, the first order error*

$$G = \sum_i e^{\beta \mu_i} \beta(\mu_i - \hat{\mu}_i)$$

*satisfies*

$$\mathbb{P}\left(|G| \geq \frac{\epsilon}{2} \sum_i e^{\beta \mu_i}\right) \leq \delta/3. \tag{29}$$

*Proof.* First, defining $E_f$ as the failure event $|G| \geq \frac{\epsilon}{2} \sum_i e^{\beta \mu_i}$, note that

$$P(E_f) = P(E_f|E_2)P(E_2) + P(E_f|\overline{E_2})P(\overline{E_2})$$

implies

$$P(E_f|E_2) = \frac{P(E_f) - P(E_f|\overline{E_2})P(\overline{E_2})}{P(E_2)} \leq 2P(E_f),$$

where we use that since $\delta < 1$, $P(E_2) \geq 1/2$. Thus, it suffices to show that $P(E_f) \leq \delta/6$

G is a sum of independent sub-Gaussian random variables, each scaled by a constant. Thus, we have the two-sided tail bound that with probability at least $1 - \delta/6$

$$\left|\sum_i e^{\beta \mu_i} \beta(\mu_i - \hat{\mu}_i)\right| \leq \sqrt{2B^2 \log(12/\delta)}$$

with the sum having sub-Gaussian parameter

$$B^2 = \sum_i e^{2\beta \mu_i} \beta^2 \sigma^2/n_i$$

Plugging in our value of $B^2$ we find

$$\left| \sum_i e^{\beta \mu_i} \beta (\mu_i - \hat{\mu}_i) \right|$$

$$\leq \sqrt{2 \log(12/\delta) \sum_i e^{2\beta \mu_i} \beta^2 \sigma^2 / n_i}$$

$$\leq \beta \sigma \sqrt{2 \log(12/\delta)} \left( \sum_i e^{2\beta \mu_i} \frac{1}{\frac{e^{\beta \mu_i} T}{2 \sum_j e^{\beta \mu_i}}} \right)^{1/2}$$

$$\leq \beta \sigma \sqrt{2 \log(12/\delta)} \left( \left( 2 \sum_j e^{\beta \mu_i} \right) \sum_i e^{\beta \mu_i} \frac{1}{T} \right)^{1/2}$$

$$= 2 \beta \sigma \sqrt{\log(12/\delta)} f(\mu) / T^{\frac{1}{2}},$$

where in (30) we use that on $E_1$, $\hat{\alpha}_i \geq \frac{1}{2} \alpha_i$, and so $n_i \geq \alpha_i T/2$. Thus

$$T \geq 16 \beta^2 \sigma^2 \epsilon^{-2} \log(12/\delta) \tag{30}$$

is sufficient to yield the desired multiplicative error of $\varepsilon/2$ with probability at least $1 - \delta/3$. $\qquad\square$

**Lemma 6.** *If arm $i$ is pulled at least $\frac{4\sqrt{2}\sigma^2 \beta^2 \log(6n/\delta)\gamma_i \sum_{j=1}^n \gamma_j}{\epsilon \sum_j \alpha_j}$ times, then on the success events $E_1, E_2$ where the confidence intervals hold,*

$$\sum_{j=1}^n \beta^2 e^{\beta \mu_j} (\mu_j - \hat{\mu}_j)^2 \leq \frac{\varepsilon}{2} \sum_j e^{\beta \mu_j} \tag{31}$$

*and Algorithm 2 will require a number of arm pulls at most*

$$\frac{91 \sigma^2 \beta^2 \log(6n/\delta) \left( \sum_{i=1}^n \gamma_i \right)^2}{\epsilon \sum_i \alpha_i} \tag{32}$$

*Proof.* Bounding the second order error, we utilize the fact that we have sampled proportional to $\hat{\gamma}_i$. On the event $E_2$ (where $n_i \geq \frac{\gamma_i}{\sqrt{2}} T$) the second order error can be bounded as

$$\sum_{i=1}^n \beta^2 e^{\beta \mu_i} (\mu_i - \hat{\mu}_i)^2 \tag{33}$$

$$\leq \sum_{i=1}^n 2\sigma^2 \beta^2 e^{\beta \mu_i} \log(6n/\delta) / n_i \tag{34}$$

$$\leq 2\sigma^2 \beta^2 \log(6n/\delta) \sum_{i=1}^n \frac{e^{\beta \mu_i}}{\hat{\gamma}_i T} \tag{35}$$

$$\leq \frac{2\sqrt{2}\sigma^2 \beta^2 \log(6n/\delta)}{T} \sum_{i=1}^n \frac{e^{\beta \mu_i}}{\gamma_i} \tag{36}$$

$$= \frac{2\sqrt{2}\sigma^2 \beta^2 \log(6n/\delta)}{T} \left( \sum_{i=1}^n e^{\beta \mu_i / 2} \right)^2 \tag{37}$$

We want this second order error to be at most $\frac{\epsilon}{2} \sum_i e^{\beta \mu_i}$, and so require $T$ to satisfy the inequality below

$$T \geq \frac{4\sqrt{2}\sigma^2 \beta^2 \log(6n/\delta) \left( \sum_{i=1}^n e^{\beta \mu_i / 2} \right)^2}{\epsilon \sum_i e^{\beta \mu_i}}$$

$$= \frac{4\sqrt{2}\sigma^2 \beta^2 \log(6n/\delta) \left( \sum_{i=1}^n \gamma_i \right)^2}{\epsilon \sum_i \alpha_i} \tag{38}$$

Algorithmically this is not a valid $T$ to use, since it depends on the unknown $\mu_i$. However, since $\hat{\alpha}_i, \hat{\gamma}_i$ are close to their true values on the good event $E_1$, we can use these estimates. Thus, we take as our budget $T$ for this second order error:

$$T = \frac{16\sqrt{2}\sigma^2\beta^2 \log(6n/\delta) \left(\sum_{i=1}^n \hat{\gamma}_i\right)^2}{\epsilon \sum_i \hat{\alpha}_i}, \tag{39}$$

which on the event $E_1$ is larger than (38). This is a random quantity, so to analyze the requisite sample complexity, we use the fact than on $E_1$, we can bound the quantity in (39) as

$$T \leq \frac{64\sqrt{2}\sigma^2\beta^2 \log(6n/\delta) \left(\sum_{i=1}^n \gamma_i\right)^2}{\epsilon \sum_i \alpha_i}. \tag{40}$$

Note $64\sqrt{2} < 91$ so we use this simple constant in the statement of the lemma. $\qquad\square$

This can be directly compared to the case where we only sample according to $\alpha_i$, not $\gamma_i$, which would yield a sample complexity of

$$T \geq \frac{8n\sigma^2 \log(6n/\delta)\beta^2}{\epsilon} \tag{41}$$

samples. Note that since $\left(\sum_{i=1}^n \gamma_i\right)^2 \leq n \sum_i \alpha_i$ by Cauchy-Schwarz, this is always an improvement (up to absolute constants) up to a factor of $n$.

Now we can provide a proof of the main theorem.

*Proof of Theorem 1.* We utilize the adaptive approximation subroutine with error probability $\delta' = \delta/3$ and error $\varepsilon' = \varepsilon/4$ from Lemma 6. With $1 - \delta'$ probability, it requires a number of samples at most

$$\begin{aligned} T \leq \; & 34\beta^2\sigma^2 \log(18n/\delta)n \\ & + \frac{363\sigma^2\beta^2 \log(18n/\delta) \left(\sum_{i=1}^n \gamma_i\right)^2}{\epsilon \sum_i \alpha_i} \\ & + \frac{256\beta^2\sigma^2 \log(36/\delta)}{\varepsilon^2}. \end{aligned}$$

Best arm identification is called with error probability $\delta' = \delta/3$. From Lemma 1 this requires sample complexity

$$\sum_{i=1}^n \min\left(\frac{32\sigma^2 \ln\left(\frac{12n}{\delta} \log_2^2\left(4/\Delta_i\right)\right)}{\Delta_i^2}, d\right) \tag{42}$$

We then estimate the best arms mean using Lemma 2 to accuracy $\varepsilon' = \varepsilon/4$ with error probability $\delta' = \delta/3$. This requires a number of samples at most

$$\frac{512\sigma^2\beta^2 \log(6/\delta)}{\epsilon^2}. \tag{43}$$

By a union bound, all these algorithms succeed with probability at least $1 - \delta$. Analyzing the multiplicative error, we have that $[1/(1+\varepsilon'), 1/(1-\varepsilon')] \in [1 - 2\varepsilon, 1 + 2\varepsilon]$ for $0 < \varepsilon \leq 1/2$, and that $(1 \pm \varepsilon_1)(1 \pm \varepsilon_2) \in (1 \pm (\varepsilon_1 + \varepsilon_2 + \varepsilon_1\varepsilon_2))$. Thus, on these success events, the numerator is approximated to accuracy $\varepsilon/4$, and the denominator to accuracy $\varepsilon/4$. The denominator error converts to a multiplicative error of $\varepsilon/2$ in the numerator, which combines to yield an error of $3\varepsilon/4 + \varepsilon^2/8 < \varepsilon$, as $\varepsilon < .5$. This allows us to simplify the total sample complexity as

$$\begin{aligned} T &\leq \frac{512\sigma^2\beta^2 \log(6/\delta)}{\epsilon^2} + \frac{256\beta^2\sigma^2 \log(36/\delta)}{\varepsilon^2} \\ &\leq \frac{768\sigma^2\beta^2 \log(36/\delta)}{\varepsilon^2} \end{aligned}$$

Thus

$$
\begin{aligned}
T \leq & 34\beta^2\sigma^2 \log(18n/\delta)n \\
& + \sum_{i=1}^{n} \min\left( \frac{32\sigma^2\beta^2 \ln\left(\frac{12n}{\delta}\log_2^2\left(4/\Delta_i\right)\right)}{\Delta_i^2}, d \right) \\
& + \frac{363\sigma^2\beta^2 \log(18n/\delta)\left(\sum_{i=1}^{n}\gamma_i\right)^2}{\epsilon \sum_i \alpha_i} \\
& + 768\sigma^2\beta^2 \log(36/\delta)\epsilon^{-2} \\
\leq & C\left( \beta^2\sigma^2\left[ n\log\left(\frac{n}{\delta}\right) + \sum_{i=1}^{n}\min\left(\frac{\log\left(\frac{n\log d}{\delta}\right)}{\Delta_i^2}, d\right) + \frac{\log\left(\frac{n}{\delta}\right)\left(\sum_{j=1}^{n}\gamma_j\right)^2}{\epsilon \sum_j \alpha_j} + \frac{\log(1/\delta)}{\varepsilon^2}\right]\right)
\end{aligned}
$$

$\square$

Algorithmically, in the second stage, arm $i$ needs to be pulled a number of times

$$
\begin{aligned}
n_i = & 17\beta^2\sigma^2 \log(6n/\delta) \\
& + \frac{16\sqrt{2}\sigma^2\beta^2 \log(6n/\delta)\left(\sum_{i=1}^{n}\hat{\gamma}_i\right)^2}{\epsilon \sum_i \hat{\alpha}_i} \frac{\hat{\gamma}_i}{\sum_j \hat{\gamma}_j} \\
& + 16\beta^2\sigma^2 \log(12/\delta)\epsilon^{-2} \frac{\hat{\alpha}_i}{\sum_j \hat{\alpha}_j} \\
\leq & 17\beta^2\sigma^2 \log(6n/\delta) \\
& + \frac{64\sqrt{2}\sigma^2\beta^2 \log(6n/\delta)\left(\sum_j \gamma_j\right)\gamma_i}{\epsilon \sum_j \alpha_j} \\
& + \frac{16\beta^2\sigma^2 \log(12/\delta)\hat{\alpha}_i}{\epsilon^2 \sum_j \alpha_j}
\end{aligned}
$$

where we upper bound on the success event $E_2$.

Combining this with the initial $T_0$ pulls, the pulls from best arm identification, and the pulls from estimating the value of the best arm, we have that the overall sample complexity for arm $i$ is upper bounded as:

$$
\begin{aligned}
\tilde{n}_i = & 34\beta^2\sigma^2 \log(6n/\delta) \\
& + \frac{64\sqrt{2}\sigma^2\beta^2 \log(6n/\delta)\left(\sum_j \gamma_j\right)\gamma_i}{\epsilon \sum_j \alpha_j} \\
& + \frac{16\beta^2\sigma^2 \log(12/\delta)\alpha_i}{\epsilon^2 \sum_j \alpha_j} \\
& + \frac{32\sigma^2 \ln\left(\frac{12n}{\delta}\log_2^2\left(4/\Delta_i\right)\right)}{\Delta_i^2} \\
& + \frac{512\sigma^2\beta^2 \log(6/\delta)}{\epsilon^2}\mathbb{1}\{i = 1\}
\end{aligned}
$$

And so, the total algorithmic sample complexity on the success event is

$$
T \leq \sum_i \min\left(\tilde{n}_i, d\right). \tag{44}
$$

## B.1 Interpreting the results

We work to provide a simplified (looser) bound on the sample complexity when the minimum gap is $\Delta$. The worst case sample complexity in this case is when the best arm has mean $\mu_1$, and all the rest have mean $\mu_1 - \Delta$.

This allows us to simplify the overall sample complexity as

$$T \leq C\left(\sigma^2\left[\beta^2 n \log\left(\frac{n}{\delta}\right) + \sum_{i=1}^n \frac{\log\left(\frac{n\log d}{\delta}\right)}{\Delta_i^2} + \beta^2 \frac{\log\left(\frac{n}{\delta}\right)\left(\sum_{j=1}^n \gamma_j\right)^2}{\epsilon \sum_j \alpha_j} + \beta^2 \frac{\log(1/\delta)}{\varepsilon^2}\right]\right) \quad (45)$$

$$\leq C\left(\beta^2\sigma^2 \log\left(\frac{n}{\delta}\right)\left(n + \varepsilon^{-1}\left(\frac{1 + n^2 e^{-\beta\Delta}}{1 + n e^{-\beta\Delta}}\right)\right) + \beta^2\sigma^2\varepsilon^{-2}\log(1/\delta) + n\sigma^2 \log\left(\frac{n\log d}{\delta}\right)\Delta^{-2}\right)$$

where we use the fact that for $n > 2$:

$$\frac{\left(\sum_{j=1}^n \gamma_j\right)^2}{\sum_j \alpha_j} = \frac{\left(1 + (n-1)e^{-\beta\Delta/2}\right)^2}{1 + (n-1)e^{-\beta\Delta}}$$

$$\leq 2\frac{1 + (n-1)^2 e^{-\beta\Delta}}{1 + (n-1)e^{-\beta\Delta}}$$

$$\leq C\left(\frac{1 + n^2 e^{-\beta\Delta}}{1 + n e^{-\beta\Delta}}\right).$$

Evaluating our sample complexity in (45), when $\Delta > \frac{2}{\beta}\log n$, this term is bounded by a constant, and the sample complexity can be more simply bounded as.

$$T \leq C\left(\beta^2\sigma^2 \log\left(\frac{n}{\delta}\right)\left(n + \varepsilon^{-1}\right) + \beta^2\sigma^2\varepsilon^{-2}\log(1/\delta) + n\sigma^2 \frac{\log\left(\frac{n\log d}{\delta}\right)}{\log^2 n}\right)$$

$$\leq C\beta^2\sigma^2\left(\log\left(\frac{n}{\delta}\right)\left(n + \varepsilon^{-1}\right) + \varepsilon^{-2}\log(1/\delta)\right) \quad (46)$$

In the last line we require the condition that

$$\frac{\log\left(\frac{n\log d}{\delta}\right)}{\log^2 n} \leq \beta^2 \log\left(\frac{n}{\delta}\right), \quad (47)$$

i.e. $d$ is not doubly exponential in $n$ and $\beta$ is not too small.

**Assumption 2** (large gap, moderate $\beta$, moderate $d$). *We assume that* (47) *holds, and that* $\Delta > \frac{2}{\beta}\log n$.

Under the conditions of Assumption 2, the sample complexity in Theorem 1 can be simplified as in Corollary 1.

## B.2 Asymptotic optimality of sampling frequencies

Following the approach of [8], we can show the asymptotic optimality of our sampling frequencies (sampling proportional to $\alpha_i$ for minimizing the first order error, and $\gamma_i$ for minimizing our bound on the second order error).

### B.2.1 First order frequencies $\alpha_i$

Considering a plug-in estimator $\hat{\mu}$; analyzing the first order taylor expansion of its error, we have that

$$f(\hat{\mu}) - f(\mu) = \nabla f(\mu)^\top (\hat{\mu} - \mu) + O\left(\|\hat{\mu} - \mu\|_2^2\right).$$

Thus in the high accuracy regime ($\varepsilon \to 0$), we can consider only the first order term. Assuming Gaussian noise in our arm pulls (an identical result holds for sub-Gaussian noise), the first order

error can be bounded as (assuming we use $T$ pulls, and sample arm $i$, $p_i T$ times for a probability distribution $p$:

$$\nabla f(\mu)^\top (\hat{\mu} - \mu) \sim \mathcal{N}\left(0, \sum_{i=1}^{n} \frac{\beta^2 e^{2\beta\mu_i}}{T p_i}\right)$$

Optimizing over probability distributions $p$, using Sion's minimax theorem and strong duality as in [8] gives us that

$$\boldsymbol{\alpha}^\star = \operatorname*{argmin}_p \sum_{i=1}^{n} \beta^2 \frac{e^{2\beta\mu_i}}{T p_i}$$

$$= \operatorname*{argmin}_p \max_\lambda \sum_{i=1}^{n} \frac{e^{2\beta\mu_i}}{T p_i} + \lambda\left(\sum_i p_i - 1\right)$$

utilizing strong duality, we have that

$$\frac{\partial}{\partial p_i} = -e^{2\beta\mu_i} p_i^{-2} + \lambda = 0 \quad \Longrightarrow \quad \boldsymbol{\alpha}^\star \propto e^{\beta\boldsymbol{\mu}}. \tag{48}$$

Note that in the limit as $\varepsilon \to 0$, multiplicative and additive error objectives are equivalent.

### B.2.2  Second order sampling frequencies $\gamma_i$

With the first order error term in hand, the second order error term is left to be analyzed:

$$\sum_{i=1}^{n} \beta^2 e^{\beta\mu_i} (\mu_i - \hat{\mu}_i)^2. \tag{49}$$

We minimize a bound on this given by our confidence intervals $|\mu_i - \hat{\mu}_i| \leq \frac{c}{\sqrt{n_i}}$, where $c$ is some constant. We identify what sampling distribution $p$ minimizes this second order bound, defining $n_i = p_i T$.

By a similar argument as for the first order analysis:

$$\boldsymbol{\gamma}^\star = \operatorname*{argmin}_p \sum_{i=1}^{n} \beta^2 e^{\beta\mu_i} \frac{1}{T p_i}$$

$$= \operatorname*{argmin}_p \max_\lambda \sum_{i=1}^{n} e^{\beta\mu_i} \frac{1}{p_i} + \lambda\left(\sum_i p_i - 1\right)$$

utilizing strong duality, we have that

$$\frac{\partial}{\partial p_i} = -e^{\beta\mu_i} p_i^{-2} + \lambda = 0 \quad \Longrightarrow \quad \boldsymbol{\gamma} \propto e^{\beta\boldsymbol{\mu}/2} \tag{50}$$

**Gains by sampling according to $\gamma_i$**   Sampling according to $\boldsymbol{\gamma}$ gives a second order error bounded by

$$\left(\sum_i e^{\beta\mu_i/2}\right)^2 = \|\boldsymbol{\gamma}\|_1^2, \tag{51}$$

as opposed to sampling according to $\boldsymbol{\alpha}$ which gives a second order error bounded by

$$n \sum_i e^{\beta\mu_i} = n\|\boldsymbol{\gamma}\|_2^2 \tag{52}$$

By standard norm inequalities, sampling according to $\boldsymbol{\gamma}$ is always at least as good, and up to a factor of $n$ improvement in the case where one entry in the softmax is much larger than the rest; exactly the case of interest.

## B.3 Comparison with [8]

In [8], the problem of estimating a real valued function $f$ to additive accuracy $\varepsilon$ with probability at least $1 - \delta$ is studied, under the assumption that the function has $L$-Lipschitz gradients. For the case of softmax estimation, the gradients are not Lipschitz due to the unbounded nature of the exponential. However, if we evaluate the norm of the gradient at a point $\boldsymbol{\mu}$, we obtain

$$\|\nabla f(\boldsymbol{\mu})\|^2 = \lim_{c \to 0} \max_{\|\boldsymbol{u}\| \leq c} c^{-2} \|\nabla f(\boldsymbol{\mu} + \boldsymbol{u}) - \nabla f(\boldsymbol{\mu})\|_2^2$$

$$= \lim_{c \to 0} \max_{\|\boldsymbol{u}\| \leq c} c^{-2} \sum_i \left( \beta e^{\beta(\mu_i + u_i)} - \beta e^{\beta \mu_i} \right)^2$$

$$= \lim_{c \to 0} \max_{\|\boldsymbol{u}\| \leq c} c^{-2} \beta^2 \sum_i e^{2\beta \mu_i} \left( e^{\beta u_i} - 1 \right)^2$$

$$= \beta^4 \max_i e^{2\beta \mu_i}.$$

Theorem 1 of [8] states that the number of samples required to achieve $\varepsilon$ additive error with probability at least $1 - \delta$ is

$$T = O \left( \frac{\|\nabla f(\boldsymbol{\mu})\|_1^2 \log(1/\delta)}{\varepsilon^2} + \frac{n^2 L \log(n/\delta)}{\varepsilon} \right), \tag{53}$$

where the noise variance $\sigma^2$ is assumed to be 1. Since the error in our setting is multiplicative, we are interested in $\varepsilon' = \epsilon f(\boldsymbol{\mu}) = \varepsilon \sum_i e^{\beta \mu_i}$. Additionally, $\|\nabla f(\boldsymbol{\mu})\|_1^2 = \beta^2 \left( \sum_i e^{\beta \mu_i} \right)^2$. Thus, the number of samples required is

$$T = O \left( \frac{\|\nabla f(\boldsymbol{\mu})\|_1^2 \log(1/\delta)}{\varepsilon^2} + \frac{n^2 L \log(n/\delta)}{\varepsilon} \right)$$

$$= O \left( \frac{\beta^2 \left( \sum_i e^{\beta \mu_i} \right)^2 \log(1/\delta)}{\varepsilon^2 \left( \sum_i e^{\beta \mu_i} \right)^2} + \frac{n^2 \beta^4 \max_i e^{2\beta \mu_i} \log(n/\delta)}{\varepsilon \sum_i e^{\beta \mu_i}} \right)$$

$$= O \left( \frac{\beta^2 \log(1/\delta)}{\varepsilon^2} + \frac{n^2 \beta^4 \max_i e^{2\beta \mu_i} \log(n/\delta)}{\varepsilon \sum_i e^{\beta \mu_i}} \right)$$

This is to be compared with the sample complexity of the proposed algorithm in this paper, for the specific setting of softmax normalization estimation, which is

$$T \leq C\beta^2 \left( n \log \left( \frac{n}{\delta} \right) + \left( \log \left( \frac{n}{\delta} \right) \left( \sum_{j=1}^n \gamma_j \right) \right)^2 \left( \epsilon \sum_j \alpha_j \right)^{-1} + \frac{\log \left( \frac{1}{\delta} \right)}{\varepsilon^2} \right),$$

taking $\sigma^2 = 1$ to compare results.

The constant term independent of $\varepsilon$ is to linearize the exponential. The $\varepsilon^{-2}$ term matches between the two settings, as asymptotically the optimal strategy is indeed to sample according to the first derivative. The term scaling with $\varepsilon^{-1}$ improves dramatically on that of prior work. Note that in the case of $f(\boldsymbol{\mu}) = \|\boldsymbol{\mu}\|_2^2$, the second order term (scaling with $\varepsilon^{-1}$) can be improved to $O(n^{3/2} L \varepsilon^{-1})$, as the mean of the second order error can be removed. Thus, we can see the massive improvement afforded by our more refined algorithm, tailored for the specific structure of the softmax function.

## B.4 Extension to heterogeneous arm variances

Adapting bandit algorithms to settings with heterogeneous variances has been done in both the standard regret [3] and best arm identification [33] settings.

For best arm identification, sacrificing log factors for the sake of clarity, empirical-Bernstein-based confidence intervals [35] can be constructed where we iteratively pull each arm once, try and

eliminate, and progress. Union bounding over the $nd$ possible pulls naively upper bounds this, yielding a complexity of

$$O\left(\sum_{i=1}^{n}\min\left(\left(\frac{\sigma_i^2}{\Delta_i^2}+\frac{1}{\Delta_i^2}\right)\log\left(\frac{\delta}{nd}\right),d\right)\right). \tag{54}$$

This assumes that all arms are bounded in $[0,1]$ with variance $\sigma_i^2$.

We additionally require a lemma for estimating the mean of the best-arm in a PAC sense.

**Lemma 7** (Exponential best arm estimation)**.** *Sampling arm $i$*

$$T=\frac{32\sigma_i^2\beta^2\log(2/\delta)}{\epsilon^2}$$

*samples guarantees that $e^{\beta\hat{\mu}_i}$ estimates $e^{\beta\mu_i}$ to multiplicative accuracy $\epsilon$, with probability at least $1-\delta$.*

This trivially follows from the proof of Lemma 2.

For the softmax normalization estimation, we know from [8] that the optimal first order sampling frequencies are to sample arm $i$ a number of times proportional to

$$n_i \propto \sigma_i e^{\beta\mu_i}. \tag{55}$$

However, sampling like this yields an additive first order that scales as $\sum_i \sigma_i e^{\beta\mu_i}$, which cannot be easily related to $\sum_i e^{\beta\mu_i}$, as we would need to get multiplicative error bounds. Thus, we instead use suboptimal target first order sampling frequencies, scaling with $\sigma_i^2$, to avoid this analysis issue

$$n_i \propto \sigma_i^2 e^{\beta\mu_i}. \tag{56}$$

Scaling the number of pulls for each arm by $\sigma_i^2$ yields:

**Proposition 3** (Softmax normalization estimation: heterogeneous variances variant of Proposition 1)**.** *Under Assumption 1, Algorithm 2 will, with probability at least $1-\delta$, estimate $f_\beta(\boldsymbol{\mu})=\sum_j e^{\beta\mu_j}$ to a multiplicative accuracy of $\epsilon$, using a number of samples for arm $i$ at most*

$$n_i = 34\beta^2\sigma_i^2\log(6n/\delta)n + \frac{91\sigma_i^2\beta^2\log(6n/\delta)\left(\sum_{i=1}^{n}\gamma_i\right)^2}{\epsilon\sum_i\alpha_i} + \frac{16\beta^2\sigma_i^2\log(12/\delta)}{\epsilon^2}$$

The proof of this proposition follows similarly to Proposition 2, as sampling proportional to $\sigma_i^2$ cancels the differing variances, essentially reducing the problem to the homogeneous setting (suboptimally).

## C Implementation Details

In this appendix, we present the implementation details of our algorithm. In Algorithm 4, we provide pseudocode with greater detail about our implementation of the `Adaptive Softmax` algorithm. We note that Algorithm 4 contains some implementation differences from the original Algorithm 1 presented in Section 4. None of these changes materially affect the output of the algorithm; nonetheless, we provide a discussion of them here to enable reproducibility of our experimental results. Our results are also reproducible via a 1-line script in our code submission. In the following subsections, we describe each of these implementation details.

As an important note, we consider all variables global unless stated otherwise. This is to say, calling "pull arms" updates the state of arm mean estimates $\hat{\mu}$, their variance estimates $\hat{\sigma}$, the number of pulls per arm $\{n_i\}$, inclusion probabilities $\pi$, etc. We define the estimators based on Gumbel sampling according to importance weights as the set of arms $\mathcal{A}$. This idea of treating an arm simply as a sequence of estimators with confidence intervals was pioneered for the computational setting in [4], and saw further usage in [24].

### C.1 Reusing Arm Pulls

In our theoretical analysis in Appendix B, each phase of Algorithm 1 is handled independently. This allows us to union bound the error probabilities of each phase of the algorithm. In our implementation,

**Algorithm 4** `Adaptive Softmax` (implementation details)

---

1: **Input:** Matrix $A$, vector $x$, temperature $\beta$, error $\epsilon$, failure probability $\delta$
2: **Output:** With probability at least $1 - \delta$, the argmax coordinate $i^*$ and an estimate $\hat{p}_{i^*}$ of its probability such that $(1 - \epsilon)p_{i^*} \leq \hat{p}_{i^*} \leq (1 + \epsilon)p_{i^*}$.
3: $w \leftarrow \text{GetImportanceWeights}(A, x)$        ▷ Algorithm 5
4: $P, c \leftarrow \text{GumbelPermutation}(w)$
5: Construct set of arms $\mathcal{A}$ from $A, x, P, c$
6: $\sigma^2 = \frac{1}{\beta}$        ▷ Initial variance to pull arms to
7: $\text{PullToVariance}(\mathcal{A}, \sigma^2)$        ▷ Algorithm 8
8: $i^* \leftarrow \text{BestArmId}(\mathcal{A}, \delta/2, \hat{\sigma}^2)$        ▷ Algorithm 3
9: $\hat{\mu}_{i^*} \leftarrow \langle A_{i\cdot}, x \rangle$        ▷ Exact computation of $\hat{\mu}_{i^*} = \mu_{i^*}$
10: $\hat{Z} \leftarrow \text{NormalizationEstimation}(\mathcal{A}, \varepsilon/2, \delta/2, \hat{\sigma})$        ▷ Algorithm 2
11: Compute estimated probability as $\hat{p}_{i^*} = e^{\beta\hat{\mu}_{i^*}}/\hat{Z}$
12: **return** $i^*, \hat{p}_{i^*}$

---

**Algorithm 5** `GetImportanceWeights`

---

1: **Input:** Matrix $A \in \mathbb{R}^{n \times d}$, vector $x \in \mathbb{R}^d$
2: **Output:** $w \in \mathbb{R}^d$, vector of importance weights
3: **for all** $j = 1, \ldots, d$ **do**
4:      Compute $w_j = |x_j| \sum_{i=1}^n |A_{i,j}|$        ▷ $\ell_1$ norm of $i$th column of $A$
5: **end for**
6: **return** $w$        ▷ Element-wise multiplication of $\ell_1$ norms of columns of $A$, and $|x|$

---

**Algorithm 6** GumbelPermutation

---

1: **Input:** Importance weights $w$
2: **Output:** Permutation $P$, cached outputs $c$ for inclusion probability calculation

3: Draw each $\xi_i \overset{\text{i.i.d}}{\sim} \text{Gumbel}(0, 1)$
4: $L \leftarrow \log w$        ▷ Log importance weights
5: $L' \leftarrow L + \xi$        ▷ Log importance weights perturbed by i.i.d. Gumbel noise
6: $h \leftarrow \text{sorted}(L')$        ▷ Compute thresholds in decreasing order
7: $P \leftarrow \text{ordering}(L')$        ▷ Compute sorting order of thresholds (argsort)
8: $c \leftarrow (L, h)$        ▷ Store log importance weight and sorted thresholds for later use
9: **return** $P, c$

---

**Algorithm 7** InclusionProbabilities

---

1: **Input:** Sample size $k$, cached outputs $c = (L, h)$ from GumbelPermutation
2: **Output:** Inclusion probabilities vector $\pi$
3: $L, h \leftarrow c$
4: $H = -\infty$        ▷ Initialize cutoff threshold
5: **if** $k < d$ **then**
6:      $H \leftarrow h_k$        ▷ Set cutoff threshold to $k$th largest perturbed log importance weight
7: **end if**
8: $\pi = 1 - \exp(-\exp(L - H))$        ▷ Inclusion probabilities; derived from Gumbel CDF
9: **return** $\pi$

---

**Algorithm 8** `PullToVariance`

---

1: **Input:** Set of arms (sequence of estimators) $\mathcal{A}$, variance $\sigma^2$
2: Set $\zeta = .1$, multiplicative pull increase factor
3: **while** there exists an arm $i$ with $\hat{\sigma}_i^2 > \sigma^2$ **do**
4:      $\mathcal{A}' \leftarrow \{i \in \mathcal{A} : \hat{\sigma}_i^2 > \sigma^2\}$, $n_i$ is corresponding number of pulls
5:      $\text{PullArms}(\mathcal{A}', (1 + \zeta)n_i)$
6: **end while**

---

---

**Algorithm 9** PullArms

---
1: **Input:** Set of arms (sequence of estimators) $\mathcal{A}$, target number of pulls per arm $\{N_i\}$
2: **for all** arms $i$ **do**
3:     **if** $n_i \geq N_i$ **then**
4:         **continue**                 $\triangleright$ Do not pull arm $i$ if it has already been pulled $N_i$ times
5:     **end if**
6:     Compute $k_i = N_i - n_i$
7:     $\pi' \leftarrow \texttt{InclusionProbabilities}(k_i, c)$
8:     Sample $k_i$ coordinates without replacement according to weights $\pi'$
9:     Update mean estimate $\hat{\mu}_i \leftarrow \frac{n_i}{N_i}\hat{\mu}_i + \frac{k_i}{N_i}\sum_{s=1}^{k_i} \frac{X_{i,s}}{\pi'_s}$
10:    Update variance estimate $\hat{\sigma}_i^2 \leftarrow \frac{n_i^2}{N_i^2}\hat{\sigma}_i^2 + \frac{1}{N_i^2}\sum_{s=1}^{k_i}\left(A_{i,(s)}x_{(s)}\right)^2 \frac{1-\pi'_{(s)}}{\pi'^2_{(s)}}$
11: **end for**

---

---

**Algorithm 10** `EstimateArm`

---
1: **Input:** arm $i \in \mathcal{A}$, pull variance $\sigma_i^2$, multiplicative error $\epsilon$, failure probability $\delta$
2: **Output:** Mean estimate $\hat{\mu}_i$
3: $\texttt{targetVar} \leftarrow \frac{\varepsilon^2}{2\sigma_i^2 \beta^2 \log(2/\delta)}$
4: `PullToVariance`($\{i\}$, targetVar)
5: **return** $\hat{\mu}_i$

---

however, we re-use arm pulls across different parts of Algorithm 1. Intuitively, once an arm return has been observed (corresponding to a scalar multiplication of an element of $A$ with an element of $x$), it can be used to warm-start estimates of $\hat{\mu}$ and $\hat{\sigma}^2$ in later stages of the algorithm. In practice, we observe that the re-use of arm pulls does not affect the correctness of our algorithm and yields significant sample complexity improvements compared to cold-starting each stage of the algorithm independently.

## C.2 Exact Computation of Best Arm

In Line 9 of Algorithm 4, we compute the mean of our estimated best arm $i^*$ exactly, and set the "estimate" $\hat{\mu}_i = \mu_i$. This allows us to reduce the approximation fidelity required by `NormalizationEstimation` (Algorithm 2) to $\frac{\epsilon}{2}$ instead of $\frac{\epsilon}{4}$ and saves a constant factor in sample complexity. Furthermore, this computation of $\mu_i$ is efficiently computed as a vector-vector dot product.

In practice, we found that Algorithm 1 usually required over $d$ samples for the best arm. As such, performing the computation $\mu_i = \langle A_{i\cdot}, x \rangle$ (reusing coordinate-wise samples from previous stages of the algorithm) did not significantly increase sample complexity.

## C.3 Initial Pulls ($T_0$)

In `NormalizationEstimation` (Algorithm 2), the initial number of arm pulls $T_0$ depends on $\sigma^2$. However, as discussed in Appendix A.3, this variance proxy is often unknown *a priori*. As such, we set $T_0 = \frac{d}{10}$. We observe that this choice of $T_0$ works well in experiments, across multiple datasets.

## C.4 Tuning

We note that despite the changes made above, there still exists some looseness in practice. To remedy this, we scale our variance estimates by a constant factor to reduce the amount of pulls needed to reach target variances, and improving the gains of our algorithm as a result. We generate these constant factors for each dataset/model by tuning on a separate "training" group of queries. Tuning is performed, generally, via bisection to discover the minimal factor which still satisfies our provided failure probability parameter $\delta$. This bisection is performed in geometric space and terminates when the $\log_{10}$ difference between the low and high end of our interval is within $10^{-2}$. The range of factors we consider is $[10^{-6}, 1]$. We first tune the constant factor independently for the variance estimates

used in bandits such that the algorithm successfully identifies the best arm with a rate of at least $1 - \delta$ on our training set. Next, we tune the constant factor for the variance estimate used in log norm estimation such that the *entire SFTM algorithm* succeeds with a rate of at least $1 - \delta$ on our training set.

## C.5  Wall-clock improvement

The focus of this paper is to develop the first provably adaptive softmax algorithm with PAC guarantees, highlighting its dramatically improved sample complexity across a wide variety of models and tasks. The eventual goal of this method is to enable wall-clock improvements in hardware implementations. These next steps of converting our provably and empirically performant method into a hardware optimized wall-clock efficient algorithm is an exciting direction of future work, which we detail below. In most modern-day transformer architectures, memory I/O serves as the primary bottleneck [20]. AdaptiveSoftmax already presents an opportunity to significantly scale down the number of entries of the matrix that must be loaded at inference time, and, in the future - if memory remains the bottleneck - improve model bandwidth by a similar factor. This objective appears in reach, since we have designed the components of AdaptiveSoftmax to be amenable to tiling and parallelization. Most notably, our implementation of AdaptiveSoftmax uses the same column to generate an importance-weighted sample for each active arm. The reasons for this implementation decision are two-fold. First, it takes advantage of the locality of entries in the same column to load samples faster, and, second, it removes intra-column correlation, which can yield theoretically improved performance [7]. Adjacent column samples can also be combined by simply summing their respective importance weights - admitting a simple tiling of our matrix that could easily be sized particularly to fit individual tiles into SRAM on a GPU along with a copy of the vector and the current mean/variance estimates for each arm. Then, we can dynamically load these tiles into SRAM based on the arm estimates as we do currently. The successive elimination bandits algorithm utilized by AdaptiveSoftmax is also, by choice, quite easily parallelizable. We may also store two copies of our matrix — one with wider tiles and one with taller tiles — to take advantage of this structure at all stages of the AdaptiveSoftmax algorithm: both when a larger number of samples is necessary for fewer arms, in later stages of adaptive estimation, and when a smaller number of samples is necessary for many arms, in earlier stages of the adaptive estimation. This said, we observe in our experiments that the bulk of compute is invested in our early samples of many arms. Just using basic parallelization to speed up this step could result in the desired speed improvements.

## C.6  Robustness to parameters

The user-desired parameters can take a wide range. We kept $\epsilon = 30\%$ constant across all simulations because we observed varying $\epsilon$ did not result in significant changes to the performance of adaSoftmax. Further, we suspect that for most users, $\delta$ will fall in the range we considered: $90 - 99\%$. However, to assuage any concerns and verify our assertion that adaSoftmax is not sensitive to choice of $\epsilon$ or $\delta$, we include here adaSoftmax with a much wider range of parameters on the MNIST dataset in Table 3.

|  | $\varepsilon = 0.001$ | $\varepsilon = 0.01$ | $\varepsilon = 0.1$ | $\varepsilon = 1.0$ |
|---|---|---|---|---|
| $\delta = 0.01$ | 100%, 5.18x | 100%, 6.63x | 99.38%, 8.13x | 99.38%, 8.14x |
| $\delta = 0.05$ | 99.75%, 6.64x | 99.13%, 8.46x | 95.38%, 8.80x | 93.50%, 8.81x |
| $\delta = 0.2$ | 91.25%, 7.19x | 89.25%, 8.90x | 88.00%, 9.29x | 83.375%, 9.25x |

Table 3: Success rate (%) and FLOP gains (x) for adaSoftmax with varied $\delta$ and $\varepsilon$ on the MNIST dataset, showing the improved performance across a wide range of parameters, and that raising $\varepsilon$ past 0.1 causes minimal difference in performance.

# D  Additional extensions and comments

## D.1  Effect of temperature

Temperature is treated as a fixed constant (fixed parameter for the problem at hand, not tunable by the algorithm). This is because tuning the temperature fundamentally changes the problem. With higher temperatures, the only arms that matter are the best and second best arms, and so adaptivity

is extremely helpful. At low temperatures, the output will be essentially the uniform distribution, and the computation is trivial and adaptivity unhelpful. With respect to other parameters, the error probability and FLOP gains of adaSoftmax are insensitive to changes in $\varepsilon$ and vary most with the choice of $\delta$. We demonstrate this trend on the MNIST dataset in Table 3.

### D.2 Application to Nucleus Sampling

The analysis posed in this paper focuses on identifying the largest entry in the softmax output and estimating its associated probability. As discussed above, this naturally extends to identifying the $k$ largest elements in the output vector by replacing the bandit best arm identification algorithm with any top-$k$ identification algorithm [39]. However, in LLM inference, the goal is often to draw a sample from the softmax output distribution via nucleus sampling [17]. Nucleus sampling avoids specifying $k$ directly; instead, it provides cumulative probability $p$ and requires the identification of the top $k$ elements such that $k$ is the smallest value such that the sum of the probabilities of the top $k$ elements is greater than $p$. The next token is then sampled from the renormalization probability distribution on these $k$ elements. Our adaptive sampling algorithm naturally applies to the nucleus sampling setting. AdaptiveSoftmax can maintain a predicted set of arms $S$ such that the sum of the arm probabilities $\hat{p}$ is greater than $p$ based on pessimistic arm mean estimates. Then, we iteratively sample: a) arms in $S$, sampling both the arm with the lowest mean minus LCB (in an attempt to verify the boundary), as well as the arm with the widest confidence interval (in order to better estimate $\hat{p}$), and b) sampling the top arm in $[n] \setminus S$, to see if it belongs in $S$. For simplicity and concreteness, in this work we focus on identifying and estimating the probability of the top-1 element, but this is an exciting direction of future work.

