# OpenReview forum: "Adaptive Sampling for Efficient Softmax Approximation"
_NeurIPS.cc/2024/Conference — NeurIPS 2024 poster_

### Official Review · Reviewer_wft8 · 2024-06-14

**Soundness:** 2
**Presentation:** 2
**Contribution:** 2
**Rating:** 5
**Confidence:** 1

**Summary:**

This paper introduces an efficient algorithm called AdaptiveSoftmax, designed to compute the top k outputs of the softmax function more effectively than the traditional full softmax computation. The key innovation lies in its adaptive approach, which leverages multi-armed bandit techniques to prioritize computations for the largest input elements, significantly reducing sample complexity. The paper provides PAC (Probably Approximately Correct) guarantees for AdaptiveSoftmax, demonstrating its theoretical soundness. Empirical results on both real and synthetic datasets, including the EuroSAT dataset, validate the algorithm's efficiency, showing substantial reductions in computational overhead, often achieving tenfold improvements or more. Additionally, the proposed method for estimating the softmax partition function offers potential applications beyond the current scope, such as in kernel density estimation.

**Strengths:**

1. The Adaptive Softmax algorithm significantly reduces computational overhead compared to the full Softmax computation. This is particularly beneficial in high-dimensional settings, making the approach practical for large-scale Machine Learning applications. The concept of focusing on the relevant top-k outputs is interesting.

2. The paper provides strong theoretical foundations with PAC guarantees for the AdaptiveSoftmax algorithm. These guarantees ensure that the method is not only empirically effective but also theoretically sound, offering reliable performance bounds. There have been extensive experiments on real and synthetic data. The authors also provide results on a variety of networks, as CNNs and LLMs.

**Weaknesses:**

1. The method relies on the assumption of a variance proxy bound for the sub-Gaussian parameters of the constructed estimators. While the paper discusses loosening this assumption, its practical implications and the extent to which it holds in various scenarios are not thoroughly explored, potentially limiting the algorithm's applicability in more varied or less controlled environments.

2. The paper does not compare quantitatively to other adaptive Softmax methods, like A

[A] Joulin, Armand, Moustapha Cissé, David Grangier, and Hervé Jégou. "Efficient softmax approximation for GPUs." In International conference on machine learning, pp. 1302-1310. PMLR, 2017.

**Questions:**

1. Please discuss if the assumptions are a constraint for the environment, does it have an impact on the effectiveness of the algorithm?
2. It would be nice to see some quantitative comparison of the proposed method with other existing algorithms, eg., speed improvements.

**Limitations:**

The authors have provided an extensive explanation of the limitations in the final section of the paper.

---

> ### Author Rebuttal · Authors · 2024-08-07
>
> We thank Reviewer wft8 for their careful reading of our discussion of empirical and theoretical improvements, and for highlighting this relevant reference. We discuss this paper, and the assumption of sub-gaussianity below.
>
> 1. **Sub-gaussian assumption:** this is a very fair question, which we discuss in detail in main response point 1. Briefly, assumptions are necessary to avoid $\Omega(nd)$ sample complexity, and empirically our assumptions are borne out. Folklore suggests that trained weights of large models are generally normally distributed, and we have verified this for the models examined in this paper (see attached Figure 1).
> 2. **Comparison:** following up on main point 2, no other methods provide theoretical guarantees (our PAC formulation in equation 4) aside from the naive baseline of exact computation, which we compare to. The method of the suggested paper "Efficient softmax approximation for GPUs" does not provide PAC guarantees, and only gives empirical wall clock speedups without any theoretical analysis of the number of samples needed, making it difficult to compare directly. At a high-level, the suggested method finds a learnable quantization (referred to as ``clusters'') of the matrix $A$, where the weights of the clusters are learned during training. Already, this poses a problem for inference since the vocabulary in the test set may be distributed differently than the training set for which the clusters are optimized. This falls into the concerns made by reviewer kEa9 of distributional shifts which our algorithm does not suffer from. We attempted to implement this method to compare against, but ran into some issues with their implementation. For example, note that the *forward* method (defined as a *torch.nn* module at [link](https://pytorch.org/docs/stable/generated/torch.nn.AdaptiveLogSoftmaxWithLoss.html) requires as input *target* which are the output labels that are not available during inference time. In comparison, our algorithm includes the cost of approximating these labels (i.e. the best-arm identification) and does so efficiently while providing PAC guarantees.

---

> > ### Comment · Reviewer_wft8 · 2024-08-08
> >
> > Thank you for the insightful discussion!

---

> > > ### Author Response · Authors · 2024-08-08
> > >
> > > Thank you for the swift response. We have incorporated yours and the other reviewers’ feedback and believe it has improved the quality of our paper. Further, it appears that the rebuttal has properly addressed the points raised in your review. If there are no further concerns, we would appreciate if your score were updated to reflect this. Please let us know if there is any additional information we can provide to help clarify.

---

> > > > ### Author Response · Authors · 2024-08-12
> > > >
> > > > Thank you again for your detailed comments and valuable reference, which have helped improve the quality of the paper. We would like to follow up, and see if you have any remaining questions or concerns that we can address during this discussion period.

---

### Official Review · Reviewer_kEa9 · 2024-07-09

**Soundness:** 3
**Presentation:** 4
**Contribution:** 3
**Rating:** 7
**Confidence:** 2

**Summary:**

This paper focuses on the efficient approximation of the softmax function. The authors propose an algorithm named AdaptiveSoftmax, which aims to reduce the computational cost of the softmax function in high-dimensional environments. Inspired by the multi-armed bandit problem, the algorithm adaptively allocates computational resources to important output values, efficiently calculating the top k values of the softmax function. It also provides PAC (Probably Approximately Correct) guarantees, demonstrating its effectiveness both theoretically and experimentally.

**Strengths:**

Originality:

The AdaptiveSoftmax algorithm introduces a novel approach by adaptively allocating computational resources, addressing challenges in existing softmax approximation methods. Traditional approaches, such as using hierarchical models, reduce computational complexity but increase the number of internal classifiers, lacking accuracy guarantees. Methods leveraging label prior distributions require prior knowledge, limiting their applicability.

Quality and Presentation:

The paper provides detailed theoretical analysis and experimental validation, robustly supporting its technical claims. The presentation is clear, with a well-organized narrative from the review of related work to the proposal of the algorithm, its theoretical guarantees, and experimental results.

Significance:

The paper offers a novel solution to a significant problem in computing the softmax function. The AdaptiveSoftmax algorithm not only substantially improves computational efficiency compared to existing methods but also provides unique PAC guarantees. This algorithm holds potential for significant computational efficiency improvements in machine learning models dealing with high-dimensional data, potentially impacting a wide range of applications.

**Weaknesses:**

In practical implementation, parameter tuning may be necessary. An ablation study on the sensitivity and justification of default hyperparameters would enhance the paper. Additionally, discussing scenarios where the proposed method may not be suitable would strengthen the paper. For instance, while the performance degradation due to approximation is shown to be limited on average, what about the worst-case scenarios, such as cases with significant data distributional shift?

**Questions:**

How was the performance of AdaptiveSoftmax evaluated? For example, are the speedup metrics in Tables 1 and 2 theoretical values or execution time comparisons? If it is the latter, what benchmarks were used? PyTorch’s torch.nn.CrossEntropyLoss utilizes log_softmax and nll_loss, which are highly optimized for GPU performance. Has AdaptiveSoftmax been compared with these in practical settings on GPUs?

**Limitations:**

The authors appropriately mention the limitations of the proposed algorithm, noting that it is most beneficial in high-dimensional settings. However, in practical settings involving LLMs, methods highly optimized for GPUs are usually used. Therefore, the question remains as to whether a fair comparison has been made regarding the current effectiveness of the proposed method in this setting.

---

> ### Author Rebuttal · Authors · 2024-08-07
>
> We thank Reviewer kEa9 for their in depth review: we wholeheartedly agree that this method of adaptive computation holds the potential for significant computational improvements across a wide range of applications. We respond to the specific questions below.
>
> 1. **Ablation study:** The theorems related to our algorithm’s sample complexity suggest minimal dependence on $\delta$, and slow growth with $\epsilon$ in the moderate $\epsilon$ regime, which is consistent with what we’ve observed experimentally. Following your suggestion, to better demonstrate and highlight this point, we have now added a table showing the sample efficiency scaling for a wide range of values of $\epsilon$ and $\delta$ in the Appendix (Rebuttal Table 1).
>
> 2. **Limitations:** As we discuss in the Limitations section of our paper, this method attains its largest gains in the high dimensional setting. However, as can be seen from our experimental results, even in smaller models like GPT2 ($d = 768$) our model exhibits over 7x improvement in sample complexity.
>
> 3. **Distribution shift:** Fortunately, our method does not suffer from these issues. It is designed to run at inference time and is truly instance-adaptive to the weights of the underlying model and the query. This is a great strength of our method, because as we discuss in main comment 2, many existing works such as the one referenced by Reviewer wft8 do in fact suffer from issues relating to distribution shift. Thank you for highlighting this point, we now add additional discussion to this end in our related works.
>
> 4. **Performance evaluation:** in this work, we use a proxy for FLOPS as our metric of comparison. This works by counting how many entries of the matrix $A$ the algorithm needs to observe (essentially, how many multiplications need to be performed). As this work focuses on providing a novel softmax approximation method with provable guarantees, with the goal of minimizing the number of entries of $A$ that need to be observed, this is the relevant metric to use to see that our algorithm is yielding the gains predicted by theory. The baseline that is compared to is brute force exact computation, which computes the entire $n \times d$ matrix vector product. Our algorithm requires many fewer samples across the wide range of settings and parameter regimes we tested.
>
> 5. **Comparisons:** The suggested methods all perform exact computation, and so are captured by the naive baseline in our comparisons. These methods are highly GPU optimized, implementing considerable hardware-conscious optimization, and so while their sample complexity is much worse than adaSoftmax, their wall clock complexity is better. Given the significant sample complexity improvement afforded by our method, we believe that improved wall clock performance is imminently possible, as we detail in Future Work. We discuss these details in points 2 and 3 of our main response.

---

> > ### Comment · Reviewer_kEa9 · 2024-08-12
> > **Official Comment by Reviewer kEa9**
> >
> > I thank the authors for the clarifications, I have no further questions. I would like to keep my positive score.

---

### Official Review · Reviewer_qUNV · 2024-07-10

**Soundness:** 3
**Presentation:** 2
**Contribution:** 3
**Rating:** 7
**Confidence:** 3

**Summary:**

The softmax function is a widely used tool, e.g., as an activation in the final layer of a classifier. Hence, cutting down on its computational costs can have a significant impact across the AIML field. This paper aims at this by introducing an adaptive variant of computing the softmax for the top $k$ values by estimating the normalization and the index of the likely highest value in the input by employing a multi-armed bandit setting. Furthermore, theoretical guarantees on the accuracy of the output of the adaptive algorithm are provided and can be controlled by an additional parameter $\delta$.

**Strengths:**

- Considering how ubiquitous the softmax function is in AIML methods, this paper promises a significant impact. The reported gains in sample efficiency within both synthetic and real-world datasets further underpin this point.
- The method has been presented with good clarity and has a good level of originality. Creating more flexible computation models is an important research endeavor when tackling the growing resource demands by deep learning methods while maintaining a desired level of accuracy, which is honored in this work by allowing for fine-granular control of the trade-off between resources and accuracy by using the target error and failure probability parameters.

**Weaknesses:**

- Overall, a more extensive evaluation with improved clarity is highly desirable. It would be important to see the effect of changing the $\epsilon$ parameter as well when conducting the evaluation, which is, to my understanding, set to a constant value of $30%$ throughout the experiments. Furthermore, reporting the effects of these changes could be made more comprehensive if presented in a plot with a wider range of choices for $\delta$, rather than just the three.
- As has been discussed in the limitations section, there can be a considerable trade-off between more involved, adaptive methods and easy-to-batch and parallelize brute-force computations. To my understanding, the experiments do not consider this trade-off, e.g., by reporting the wall-clock time next to the gains in sample efficiency.

**Questions:**

I have no questions for the authors.

**Limitations:**

The authors have addressed the limitations in a dedicated section.

---

> ### Author Rebuttal · Authors · 2024-08-07
>
> We thank Reviewer qUNV for their helpful feedback regarding improving the exposition of the improved algorithmic performance of our method.
> We respond to their two main points below:
>
> 1. **Varying parameters:**
> The user-desired parameters can take a wide range.
> We kept $\epsilon = 0.3$ (i.e. 30\%) constant across all simulations because we observed varying $\epsilon$ did not result in significant changes to the performance of AdaptiveSoftmax.
> Further, we suspect that for most users, $\delta$ will fall in the range we considered: $0.1-0.01$ (equivalently, 90-99\%).
> However, to assuage any concerns and verify our assertion that adaSoftmax is not sensitive to choice of $\epsilon$, we now run adaSoftmax with a much wider range of parameters on the MNIST dataset in Rebuttal table 1, and will add this figure to the appendix.
>
> 2. **Converting FLOP gains to wall-clock gains:**
> As discussed in main point 2, our algorithm can be made minimally batch adaptive, to improve its parallelizability.
> We further discuss in main point 3 the exciting line of future work towards converting the significant reduction in FLOPs achieved by AdaptiveSoftmax to improvements in wall clock speed.

---

> > ### Comment · Reviewer_qUNV · 2024-08-09
> >
> > Thank you for the clarifying responses to the reviews.
> >
> > With the added insights from the rebuttal, I am happy to improve my rating.

---

### Official Review · Reviewer_qEfP · 2024-07-17

**Soundness:** 3
**Presentation:** 2
**Contribution:** 3
**Rating:** 5
**Confidence:** 3

**Summary:**

This paper introduces an algorithm named AdaptiveSoftmax, designed to efficiently compute the top-k softmax values rather than the full softmax computation. This algorithm is particularly useful in high-dimensional settings where the computation of the full softmax function can be prohibitively expensive. The authors provide theoretical guarantees for the efficiency and accuracy of AdaptiveSoftmax and demonstrate its effectiveness through empirical results on real and synthetic datasets.

**Strengths:**

a)The paper presents a novel approach to approximating the softmax function using adaptive sampling.
b)The authors provide probably approximately correct (PAC) guarantees for the performance of the AdaptiveSoftmax algorithm, which strengthens the credibility of their approach.
c)The paper includes extensive empirical results demonstrating the algorithm's efficiency on both synthetic and real-world datasets, including significant reductions in sample complexity compared to brute-force softmax computation.

**Weaknesses:**

a)The algorithm relies on assumptions about the data distribution and variance proxies, which may not hold in all practical scenarios.
b)The adaptive nature of the algorithm introduces implementation complexity, particularly in balancing computational resources and ensuring efficient sampling.
c)While the algorithm is theoretically sound, its practical benefits are most pronounced in scenarios with very high-dimensional data. In lower-dimensional settings, the gains may be less significant.
d)It is better to give detailed explanation of each step in Algorithms. What is EstimateArm in Algorithm 1 step 8?
e)How sensitive is the algorithm's performance to the choice of parameters such as the temperature parameter?
f)Besides comparing with the full softmax computation, is there a detailed comparison with other approximate softmax algorithms?

Typo:
Line 5: “we present present an …” should be “we present an …
Line 135: “Our objective them becomes …” should be “Our objective becomes …”

**Questions:**

a)See weakness

**Limitations:**

8.The limitations is discussed in the paper.

---

> ### Author Rebuttal · Authors · 2024-08-07
>
> We thank Reviewer qEfP for their insightful and detailed feedback.
> We provide a point-by-point response to Reviewer qEfP’s comments below.
>
> a) **Sub-gaussian assumption:** This assumption is minimally restrictive, and is borne out in practice.
> We provide an in depth discussion in main response point 1, which we have added to the Appendix.
>
> b) **Computational efficiency of adaptivity:** our algorithm provably obtains sample complexity improvements, and as discussed in main rebuttal point 3, there are several exciting next steps to a final hardware optimized implementation that realizes these wall clock gains.
> On the theoretical side, there has been a surge of work over the last decade focused on developing multi-armed bandit algorithms with minimal rounds of adaptivity that still retain the same theoretical guarantees.
> As we show in this work, normalization estimation can be accomplished in 2 rounds of sampling, best arm identification in $\log(1/\epsilon)$ rounds, and best arm mean estimation in 1 final round.
> The round complexity of PAC best arm identification can be further improved, at the cost of worse dependence on the gaps ([13], Hillel et al.).
> Main rebuttal point 3 provides additional discussion, and many of these notes can be seen in our publicly available and easily reproducible code base (see attached files, will be made public on GitHub after the paper is published).
>
> c) **Dimensionality:** Our algorithm indeed displays its largest gains in the high dimensional regime, which is the natural regime future models will be focused on.
> As shown by our numerical experiments though, these gains are still substantial for moderate $d$.
> For example, considering GPT2 ($d=768$), one of the smaller language models in use, our approach still yields over 7x improvement in sample complexity.
>
> d) **Algorithm clarity:** Thank you for the suggestion.
> We were space-limited with the initial submission and moved the algorithm descriptions to the Appendix.
> With the added page for the camera ready version, we will add these algorithmic descriptions back into the main text.
> The function EstimateArm simply pulls the input arm to accuracy epsilon with probability at least $1-\delta$ (as in Lemma 2): we will clarify this and package the result as an algorithm in the final version.
>
> e) **Temperature:** Temperature is treated as a fixed constant (fixed parameter for the problem at hand, not tunable by the algorithm).
> This is because tuning the temperature fundamentally changes the problem.
> With higher temperatures, the only arms that matter are the best and second best arms, and so adaptivity is extremely helpful.
> At low temperatures, the output will be essentially the uniform distribution, and the computation is trivial and adaptivity unhelpful.
> We will add a brief discussion clarifying this to the Appendix.
> With respect to other parameters, the error probability and FLOP gains of AdaptiveSoftmax are insensitive to changes in $\epsilon$ and vary most with the choice of $\delta$.
> We demonstrate this trend on the MNIST dataset in Rebuttal Table 1.
>
> f) **Comparison:** As we discuss in overall point 2, there are no other approximate softmax algorithms that can provide PAC guarantees aside from the baseline of exact computation, which provides our point of comparison. We provide a qualitative comparison to a baseline without PAC guarantees suggested by another reviewer kEa9 (see **Comparison**).
>
> g) **Typo:** Thanks you for catching this, fixed.

---

> > ### Author Response · Authors · 2024-08-12
> >
> > Thank you again for the detailed review and helpful comments. In our rebuttal we worked to answer and incorporate, in a point by point manner, all the concerns and ideas that you raised. Please let us know if you have any additional questions or concerns regarding this manuscript, we would be happy to discuss them in the remaining 1.5 days of the discussion period.

---

### Author Rebuttal · Authors · 2024-08-07

We would like to thank all the reviewers for their careful reading of our manuscript.
We were pleased to see that all reviewers appreciated the novel PAC guarantees provided by this work for efficient, instance adaptive softmax computation.
We have addressed all the comments and suggestions made by the reviewers, which has helped improve the quality and clarity of the paper.
We discuss some common points of concern below, and look forward to the upcoming discussion period.

1. **Assumption of sub-Gaussianity:** This is the only assumption that we make in this paper.
It is one of the weakest assumptions possible (does not assume that the arms are Bernoulli or Gaussian), and is a common assumption in the multi-armed bandit and adaptive computation literature [4].
Unfortunately, without such an assumption, no nontrivial results are possible; consider the case where we do not have preprocessing access to $A$, the vector $x$ is all ones, and $A$ is all $1$s except for one randomly selected entry which has value $2$.
In this case, any algorithm for PAC computation of softmax$(Ax)$ with $\delta < 1-1/n$ (even just identification of the largest entry of $Ax$) requires $\Omega(nd)$ samples.
More practically though, these vectors are the result of a machine learning pipeline, and not of adversarial construction.
As shown by our simulations, this worst case scenario never occurs in practice, and arm pulls are generally well approximated by a Gaussian (see Figure 1 in the attached pdf, now added to the Appendix).
Additionally, note that for any fixed problem instance (fixed matrix $A$ and $x$), all arm pulls are bounded, and are thus sub-Gaussian.
We have added this more detailed discussion to the Appendix, and have added a reference to it in the main text.

2. **Comparison:** While many algorithms have been devised for accelerating softmax computation, no existing methods, to our knowledge, provide $(\epsilon,\delta)$-PAC guarantees (as we formulate in equation (4)) save for exact computation.
Current approximations use some combination of Locality Sensitive Hashing (LSH) to cluster the vocabulary, truncation of the tail to approximate the normalization factor, and sketching to approximate the matrix multiplication.
However, these methods are not truly instance adaptive, and fall prey to many common flaws in machine learning pipelines. For example, reviewer wft8 references a method which learns a quantization of the matrix $A$ during training [2]. The method proposed in this paper has no theoretical guarantees, requires prior knowledge of the correct output label, and only adapts to the training data, not the actual data presented at inference. Since it focuses on learning a good quantization over the training data, it can suffer from potential distributional shifts of the training and test datasets (concerns raised by reviewer kEa9, that our method avoids). See $\textbf{Comparison}$ response to Reviewer kEa9 for more details.


3. **Converting FLOP gains to wall clock gains:** The focus of this paper is to develop the first provably adaptive softmax algorithm with PAC guarantees, highlighting its dramatically improved sample complexity across a wide variety of models and tasks.
The eventual goal of this method is to enable wall-clock improvements in hardware implementations.
We provided a brief discussion of this in the Limitations and future work section (Lines 333-352), but have now added additional discussion (below) there and to the supplement.
These next steps of converting our provably and empirically performant method into a hardware optimized wall-clock efficient algorithm is an exciting direction of future work, which we detail below.
In most modern-day transformer architectures, memory I/O serves as the primary bottleneck [1].
AdaptiveSoftmax already presents an opportunity to significantly scale down the number of entries of the matrix $A$ that must be loaded at inference time, and, in the future - if memory remains the bottleneck - improve model bandwidth by a similar factor.
This objective appears in reach, since we have designed the components of AdaptiveSoftmax to be amenable to tiling and parallelization.
Most notably, our implementation of AdaptiveSoftmax uses the same column to generate an importance-weighted sample for each active arm.
The reasons for this implementation decision are two-fold.
First, it takes advantage of the locality of entries in the same column to load samples faster, and, second, it removes intra-column correlation, which can yield theoretically improved performance [Baharav and Tse 2019].
Adjacent column samples can also be combined by simply summing their respective importance weights - admitting a simple tiling of our matrix $A$ that could easily be sized particularly to fit individual tiles into SRAM on a GPU along with a copy of the vector $x$ and the current mean/variance estimates for each arm.
Then, we can dynamically load these tiles into SRAM based on the arm estimates as we do currently.
The successive elimination bandits algorithm utilized by AdaptiveSoftmax is also, by choice, quite easily parallelizable.
We may also store two copies of our matrix $A$ — one with wider tiles and one with taller tiles — to take advantage of our tiling at all stages of the AdaptiveSoftmax algorithm: both when a larger number of samples is necessary for fewer arms, in later stages of adaptive estimation, and when a smaller number of samples is necessary for many arms, in earlier stages of the adaptive estimation.
This said, we observe in our experiments that the bulk of compute is invested in our early samples of many arms.
Just using basic parallelization to speed up this step could result in the desired speed improvements.

[1] Ivanov et al. "Data movement is all you need: A case study on optimizing transformers" 2021

[2] Joulin et al. "Efficient softmax approximation for GPUs." 2017.

---

### Decision · Program_Chairs · 2024-09-25

**Decision:**

Accept (poster)

**Comment:**

Computing full softmax can be expensive. This paper proposes a new algorithm to efficiently estimate the top-k softmax values. The algorithm is inspired by the multi-armed bandit problem: it adaptively allocates resources to important output values. The work provides PAC guarantees and empirical results.

The final recommendations from the reviewers are unanimously acceptance (7, 5, 5, 7), albeit with varying degrees.

The AC agrees with the reviewers that the problem studied is important, and the proposed method is novel and theoretically sound.

One major concern from the reviewers with less favorable rating is regarding the sub-Gaussianity assumptions. The AC agrees with the authors that this assumption is not strict.

On the other hand, the AC finds that the concerns regarding more discussions and comparisons with other method to be valid (qEfP, wft8).

The topic studied is highly related to sampled softmax. From this point of view, the literature review is not complete .
The original/early papers discussing sampled softmax are not cited:

*Bengio and Senecal, Quick Training of Probabilistic Neural Nets by Importance Sampling 2003*

*Bengio and Senecal, Adaptive Importance Sampling to Accelerate Training of a Neural Probabilistic Language Model 2008*

There has been many ways of improving the sampling distribution (in the sense of importance sampling framework) since Bengio and Senegal 2008: different types of hashing methods, kernel-based sampling etc. for example the following work and the works it reviewed.

*Blanc and Rendle Adaptive Sampled Softmax with Kernel Based Sampling 2017*

The claim that “no other methods provide theoretical guarantees“ seems to be overstated. Sampled softmax comes with theoretical guarantees from importance sampling directly. See

*Rawat et al Sampled Softmax with Random Fourier Features 2019*

We therefore suggest that the authors work on a better and more comprehensive literature review and slightly tone down the claims in the final version.

Overall, the AC finds that this is an interesting work to be shared with the community with the above revisions.